# ARTICLES

# Identification of two pathways mediating protein targeting from ER to lipid droplets

Jiunn Song [1,2], Arda Mizrak[1,2], Chia-Wei Lee [1,2], Marcelo Cicconet[2], Zon Weng Lai[1,2,3], Wei-Chun Tang[1,2,4], Chieh-Han Lu[1,2], Stephanie E. Mohr[5], Robert V. Farese Jr.[1,2,6 ✉] and Tobias C. Walther [1,2,3,4,6 ✉]

Pathways localizing proteins to their sites of action are essential for eukaryotic cell organization and function. Although mechanisms of protein targeting to many organelles have been defined, how proteins, such as metabolic enzymes, target from the endoplasmic reticulum (ER) to cellular lipid droplets (LDs) is poorly understood. Here we identify two distinct pathways for ER-to-LD protein targeting: early targeting at LD formation sites during formation, and late targeting to mature LDs after their formation. Using systematic, unbiased approaches in *Drosophila* cells, we identified specific membrane-fusion machinery, including regulators, a tether and SNARE proteins, that are required for the late targeting pathway. Components of this fusion machinery localize to LD–ER interfaces and organize at ER exit sites. We identified multiple cargoes for early and late ER-to-LD targeting pathways. Our findings provide a model for how proteins target to LDs from the ER either during LD formation or by protein-catalysed formation of membrane bridges.

Although mechanisms of protein targeting to many organelles are well understood, we know little about how proteins target to the surfaces of lipid droplets (LDs). Compared with other organelles, LDs are unusual; they are bounded by a phospholipid monolayer surrounding a lipid core[1]. How do proteins target specifically to such a monolayer? This problem is important, as LDs store lipids as metabolic fuel and membrane lipid precursors[2–4], and is relevant to human diseases, as mutations of LD proteins are linked to metabolic diseases, such as fatty liver disease (for example, *PNPLA3* (refs. [5,6]) and *HSD17B13* (ref. [7])) or lipodystrophy (*PLIN1* (ref. [8]) and *PCYT1A*[9]).

Two principal pathways[10,11] mediate LD protein targeting. In one pathway, LD proteins are synthesized in the cytoplasm and directly bind to LDs, commonly via amphipathic helix motifs that adsorb to the large, persistent phospholipid packing defects of LD surfaces[12–14]. The other pathway, endoplasmic reticulum (ER)-to-LD targeting, is less well understood and is important for proteins harbouring hydrophobic segments that are initially inserted into the ER[10,15]. Cargoes of this pathway include lipid synthesis enzymes, such as long-chain acyl-CoA ligase 3 (ACSL3) and glycerol 3-phosphate acyltransferase 4 (GPAT4) (refs. [16,17]).

Because LDs form in the ER, some ER proteins could target LDs during their formation. Indeed, a small, hydrophobic hairpin sequence derived from GPAT4, called *LiveDrop*, accumulates on LDs as they form in the ER at LD assembly complexes (LDACs) consisting of seipin and accessory proteins[18,19]. Similarly, the HPos peptide, derived from ACSL3, localizes to LDs during their formation[16].

In contrast, full-length GPAT4 is excluded from forming LDs and instead targets later to mature LDs[17]. Microscopy studies performed in *Drosophila* cells suggest that late ER-to-LD targeting involves multiple physical continuities—or membrane bridges—between the ER and LDs[17,20,21]. How such bridges are formed is unknown. The Arf1/COPI vesicular trafficking machinery[22] is required for ER-to-LD targeting of proteins, such as GPAT4 (ref. [23]) and the adipose TG lipase (ATGL)[24–26], and may promote the formation of ER–LD membrane bridges[23,27], but its function in this process is uncertain.

In this Article, we sought to unravel the mechanism underlying the formation of ER–LD membrane bridges mediating the late targeting pathway. From unbiased screening in *Drosophila* cells, we identified the protein machinery mediating late ER-to-LD targeting and the cargoes of this pathway.

## Results

**Proteins access LDs from the ER at different timepoints.** *LiveDrop*, but not full-length GPAT4, accesses LDs during their formation[17,18]. To analyse the targeting kinetics of other ER-to-LD targeting proteins, we co-expressed fluorescently tagged GPAT4 and LD-associated hydrolase (LDAH), another ER-to-LD targeting cargo[28], in *Drosophila* S2R+ cells. LDAH was enriched on LDs by 30 min after induction of LD formation, whereas GPAT4 was enriched on LDs ~3 h later (Fig. 1a).

We also tested the targeting kinetics of other LD proteins that localize to the ER in the absence of LDs[29–32]. Ubxd8, a recruitment factor for the p97 segregase, targeted LDs during formation, whereas the enzymes Ldsdh1 and HSD17B11 localized to LDs at later timepoints (Fig. 1b,c). Overexpressed HSD17B11 targeted to only some LDs, suggesting additional determinants of LD targeting for this protein (Extended Data Fig. 1a,b). Thus, ER proteins appear to use different targeting pathways to access LDs: some during LD formation and others well after LDs have formed.

[1]Department of Molecular Metabolism, Harvard T.H. Chan School of Public Health, Boston, MA, USA. [2]Department of Cell Biology, Harvard Medical School, Boston, MA, USA. [3]Harvard Chan Advanced Multi-omics Platform, Harvard T.H. Chan School of Public Health, Boston, MA, USA. [4]Howard Hughes Medical Institute, Boston, MA, USA. [5]Drosophila Research and Screening Center-Biomedical Technology Research Resource (DRSC-BTRR), Department of Genetics, Harvard Medical School, Boston, MA, USA. [6]Broad Institute of Harvard and MIT, Cambridge, MA, USA. ✉e-mail: robert@hsph.harvard.edu; twalther@hsph.harvard.edu

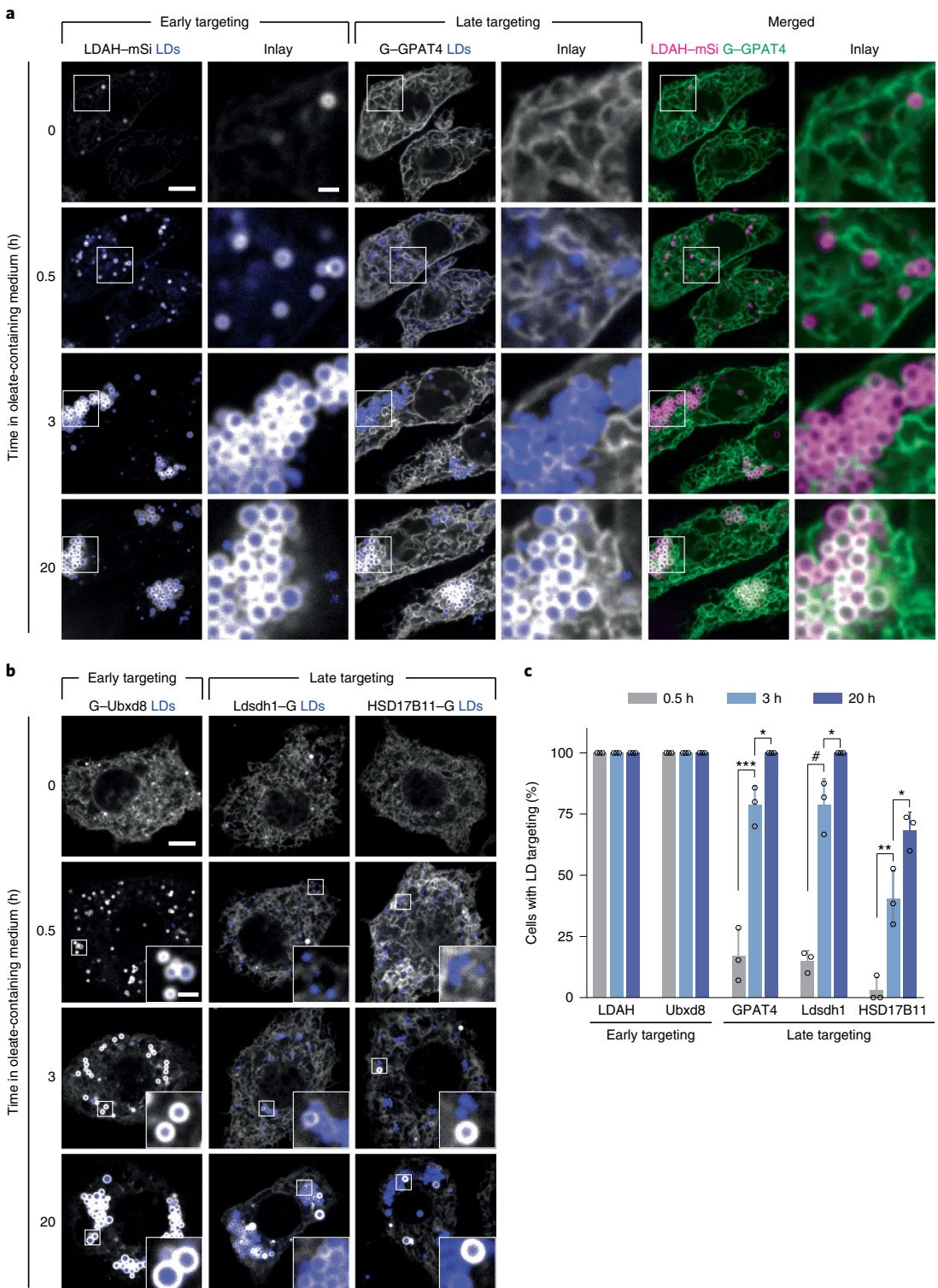

**Fig. 1 | ER proteins target LDs early during LD formation, or late after LD induction. a**, ER proteins LDAH and GPAT4 target LDs early (by 30 min) or late (after several hours), respectively, upon LD biogenesis. Confocal imaging of live *Drosophila* S2R+ cells stably overexpressing eGFP (G)–GPAT4 transfected with an LDAH–mScarlet-I (mSi) encoding construct at given timepoints after 1 mM oleic acid treatment. LDs were stained with LipidTOX Deep Red Neutral Lipid Stain. Representative images from three independent experiments are shown. Scale bars, 5 μm and 1 μm (inlay). **b**, Ubxd8 targets LDs early, and Ldsdh1 and HSD17B11 target LDs late upon LD biogenesis. Confocal imaging of live wild-type cells transiently transfected with eGFP tagged constructs at given timepoints after 1 mM oleic acid treatment. LDs were stained with monodansylpentane (MDH). Representative images are shown. Scale bars, 5 μm and 1 μm (inlay). **c**, Bar graph showing percentage of cells with LD targeting over time from the imaging experiment in **a** and **b**. For HSD17B11, cells with LD targeting were defined as those with more than two LDs with protein targeting in the imaging plane (Extended Data Fig. 1a,b). Mean ± standard deviation (s.d.), n = 3 experiments (10–16 cells each). One-way analysis of variance (ANOVA) with Bonferroni correction, *P < 0.05 (from left to right: 0.0442, 0.0240 and 0.0195), **P = 0.0048, ***P = 0.0002, #P < 0.0001. Source numerical data are available in source data.

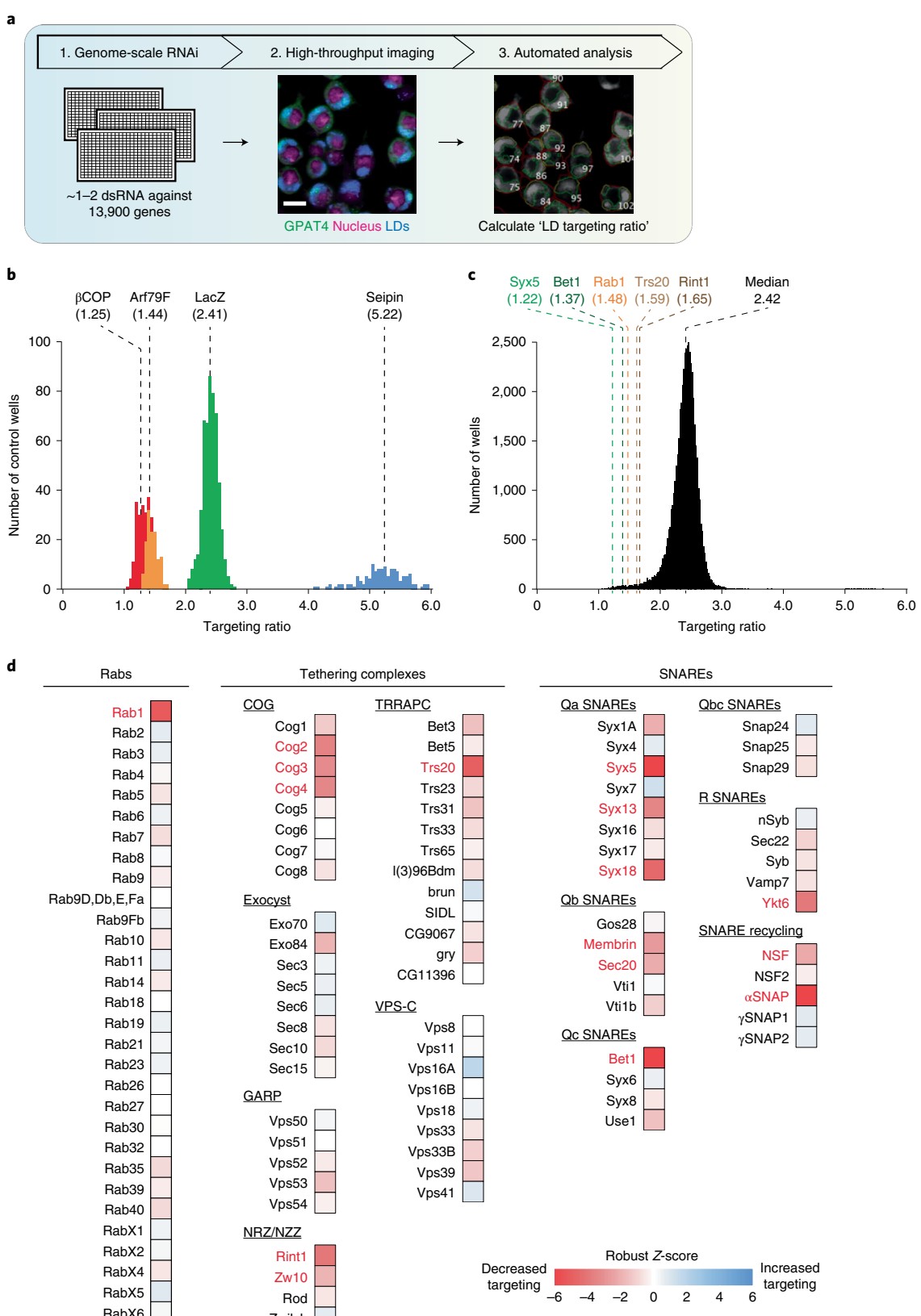

**Fig. 2 | Genome-scale imaging screen reveals that the membrane-fusion machinery is required for GPAT4 targeting to LDs. a**, Overview of genome-scale imaging screen. Scale bar, 10 μm. **b**, Histogram of LD targeting ratios of screen controls (n = 528 for LacZ; n = 132 for βCOP, Arf79F and seipin). Dotted lines indicate median values for each control. **c**, Histogram of all targeting ratios in the screen (n = 50,688). Median of all targeting ratios is indicated in black. Targeting ratios of select screen hits are also indicated. **d**, Heat map of robust Z-scores for different classes of membrane fusion machinery (Rabs, tethering complexes and SNAREs) from the imaging screen. Genes of which knockdown results in robust Z-score < −2.5 are highlighted in red. Source numerical data are available in source data.

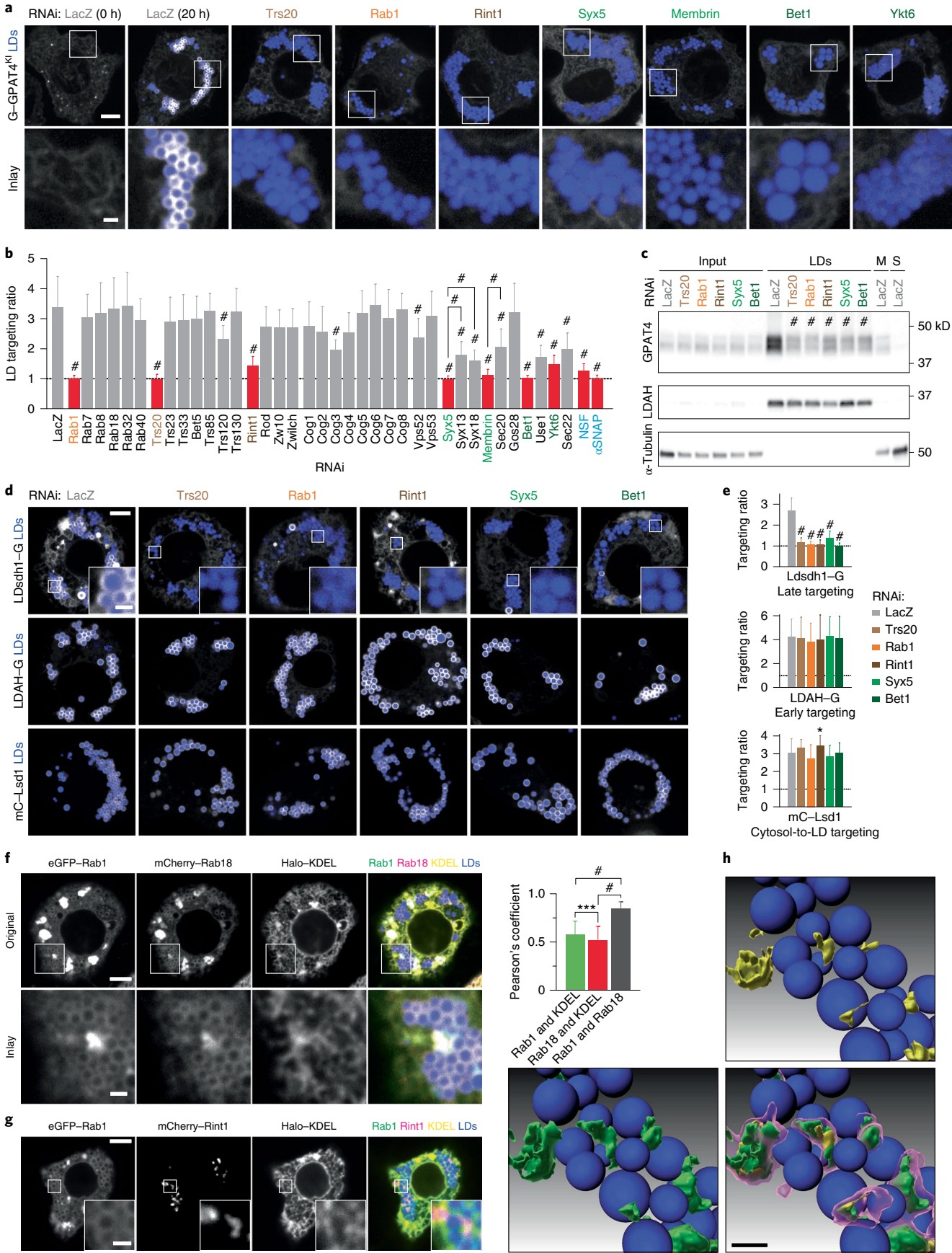

**Fig. 3 | A membrane-fusion regulator, a tether and SNAREs are required for late ER-to-LD protein targeting. a**, Depletion of specific Rab, membrane-tethering complex components and SNAREs abolished endogenous GPAT4 targeting to LDs. Confocal imaging of eGFP–GPAT4$^{KI}$ cells upon RNAi of membrane-fusion machinery components, followed by a 20-h incubation in oleate-containing medium. Scale bars, 5 μm and 1 μm (inlay).
**b**, Quantification of **a** and Extended Data Fig. 2a. Mean ± s.d., $n$ = (left to right) 59$^†$; 31$^†$, 19, 21, 45$^†$, 19 and 18; 26$^†$, 31$^†$, 32$^†$, 18, 30, 38$^†$ and 33$^†$; 20, 29, 18 and 19; 19, 20, 17, 19, 19, 20, 19 and 18; 23$^†$, 20; 27$^†$ 29$^†$ and 25$^†$; 18, 37$^†$ and 20; 43$^†$, 18; 24$^†$ and 22$^†$; 17$^†$ and 30$^†$ cells examined over two or three$^†$ independent experiments. Red: knockdowns that abolish GPAT4 targeting to LDs on imaging. One-way ANOVA with Bonferroni correction, #$P < 0.0001$, compared with LacZ unless otherwise indicated. **c**, Depletion of specific Rab, membrane-tethering complex components and SNAREs reduces GPAT4 amount in LD fractions. Western blot analysis of wild-type cell fractions upon RNAi and LD induction. Left: protein target. Right: ladder positions. M, membranes; S, soluble fraction. GPAT4 band intensities in LD fractions: LacZ (1.00), Trs20# (0.34 ± 0.06), Rab1# (0.28 ± 0.03), Rint1# (0.37 ± 0.04), Syx5# (0.35 ± 0.03) and Bet1# (0.38 ± 0.04) (mean ± s.d., $n$ = 3). One-way ANOVA with Bonferroni correction, #$P < 0.0001$ compared with LacZ.
**d**, Depletion of specific Rab, membrane-tethering complex components and SNAREs impairs LD targeting of Ldsdh1 but not of LDAH or Lsd1. Scale bars, 5 μm and 1 μm (inlay). **e**, Quantification of **d**. Mean ± s.d., $n$ = (left to right; top to bottom) 79, 48, 48, 45, 49, 45; 87, 36, 36, 39, 34, 40; 79, 36, 31, 33, 31, 33 cells examined over three independent experiments. One-way ANOVA with Bonferroni correction, *$P = 0.0469$, #$P < 0.0001$, compared with LacZ.
**f**, Localization of transiently expressed eGFP–Rab1, mCherry–Rab18, and Halo–KDEL with respect to LDs. Scale bars, 5 μm and 1 μm (inlay). Right: Pearson's correlation coefficient of intensities between two channels. Mean ± s.d., $n$ = 23 cells examined over three independent experiments. One-way ANOVA with Bonferroni correction, ***$P = 0.0001$, #$P < 0.0001$, compared with LacZ. **g**, Localization of transiently expressed eGFP–Rab1, mCherry–Rint1 and Halo–KDEL with respect to LDs. Representative images from three independent experiments are shown. Scale bars, 5 μm and 1 μm (inlay).
**h**, Three-dimensional reconstruction of images from **g**. mCherry–Rint1 puncta co-localizes with LD surface and ER. Blue: LDs; magenta: mCherry–Rint1; yellow: overlap between mCherry–Rint1 and ER (Halo–KDEL); green: overlap between mCherry–Rint1 and LD surface (eGFP–Rab1). See also Extended Data Fig. 6b,c. Scale bar, 1 μm. Source numerical data and unprocessed blots are available in source data.

### Genome-wide screen for late ER-to-LD protein targeting.

To address how cargoes target mature LDs, we systematically screened the genome for factors required for GPAT4 targeting to LDs (Fig. 2a). Specifically, we determined the effects of RNA interference (RNAi)-mediated protein depletions on LD targeting of stably expressed, fluorescently tagged GPAT4. Duplicate experiments were performed for the entire genome, collecting eight images for each knockdown and generating ~1.2 million images. Automated image analysis segmented cells and LDs to calculate an LD targeting ratio for each cell (by dividing fluorescent signal of GPAT4 on LDs by the signal outside LDs; Extended Data Fig. 1c) and the median value across all cells was reported as the readout for each knockdown (Supplementary Table 1). Plotting the distribution of LD targeting ratios across all knockdowns revealed a normal distribution with a median of 2.42, similar to control RNAi against *LacZ* (not expressed in *Drosophila*; Fig. 2b,c). Depleting most gene products had no effect on GPAT4 targeting to LDs. In contrast, depleting the positive-control proteins βCOP or Arf1 (ref. [23]) decreased LD targeting of GPAT4, whereas depleting seipin increased GPAT4 targeting[18] (Fig. 2b and Extended Data Fig. 1d). Results from replicate screens correlated well ($R = 0.7645$; Extended Data Fig. 1e). All screen images and analyses are deposited at the Lipid Droplet Knowledge Portal[33] (http://lipiddroplet.org/).

We focused on genes with robust $Z$-scores of <–2.5 or >2.5. These cut-offs yielded 910 genes that decreased and 214 genes that increased GPAT4 targeting upon knockdown out of ~13,900 genes tested, excluding ribosomal, proteasomal or spliceosomal genes. Analysis of the 910 genes showed the enrichment of protein complexes[34] involved in vesicle fusion and tethering (Extended Data Fig. 1f). Removing genes whose knockdowns resulted in extremely small LDs (which makes LD targeting ratio calculations unreliable) or significant cell death (cell count robust $Z$-score < −2.5) yielded 302 gene 'hits'. Gene Ontology analyses of the 302 genes showed enrichment of genes involved in vesicle-mediated trafficking (Supplementary Table 2).

### Membrane-fusion factors are required for late LD targeting.

Among the genes required for GPAT4 targeting, we detected a Rab protein, a membrane tether, specific SNAREs and proteins that recycle membrane-fusion machinery (Supplementary Table 3). Many common hits were found between our screen and a secretory pathway screen[35], but the overall correlation was poor, as many gene knockdowns that inhibited protein secretion did not affect GPAT4 targeting to LDs ($R = 0.3785$; Extended Data Fig. 1g).

Of the 30 *Drosophila* Rab GTPases, only Rab1 (robust $Z$-score = −5.5) was required for GPAT4 targeting to LDs (Fig. 2d). To validate the specificity of this finding, we designed two to three

**Fig. 4 | ERES organizers associate with LDs and are required for late ER-to-LD protein targeting. a**, Heat map of robust $Z$-scores for ER exit site organizers and coiled-coil tethers from the imaging screen. Red: gene knockdowns with robust $Z$-scores < −2.5. **b**, Depletion of ERES components abolishes endogenous GPAT4 targeting to LDs. Confocal imaging of eGFP–GPAT4$^{KI}$ cells upon RNAi of ERES components, followed by a 20-h incubation in oleate-containing medium. Scale bars, 5 μm and 1 μm (inlay). **c**, Quantification of **b**, including select coiled-coil tethers. Mean ± s.d., $n$ = (left to right) 59$^†$; 57$^†$, 53$^†$, 31$^†$, 67$^†$ and 37$^†$; 36, 37, 35, 35, 37, 39 and 34 cells examined over two or three$^†$ independent experiments. One-way ANOVA with Bonferroni correction, #$P < 0.0001$, compared with LacZ. **d**, Depletion of ERES components reduces GPAT4 amount in LD fractions. Western blot analysis of fractions of wild-type *Drosophila* S2R$^+$ cells upon RNAi and LD induction. Left: protein target. Right: ladder positions. Sol, soluble fraction. GPAT4 band intensities in LD fractions are indicated (mean, $n$ = 3 experiments). One-way ANOVA with Bonferroni correction, **$P < 0.01$ (from left to right: 0.0043, 0.0019), compared with LacZ. **e**, ERES components Sec16 and Tango1 associate spatially with LDs 4 h after LD induction. Immunofluorescence in wild-type cells after 1 mM oleic acid treatment. Scale bars, 5 μm and 1 μm (inlay). Bar graph shows percentages of Sec16 or Tango1 puncta associated with LDs, calculated in three-dimensional space per cell. Mean ± s.d., $n$ = (left to right; top to bottom) 39, 34 and 37; 39, 35 and 37 cells examined over two independent experiments. One-way ANOVA with Bonferroni correction, ***$P = 0.0002$, #$P < 0.0001$. **f–h**, Overexpressed Sec16 localizes around LDs and recruits endogenous Tango1 and transiently expressed Sec23 to LDs. **f** shows percentage of Tango1 or Sec23 area near LDs (defined as within one pixel distance from LDs) upon Sec16 or control overexpression (OE) from the imaging experiment in **g** and **h**. Mean ± s.d., $n$ = (left to right; top to bottom) 30, 32; 29, 32 cells examined over three independent experiments. Two-tailed Student's $t$-test, #$P < 0.0001$. **g** and **h** show confocal images of Tango1–eGFP$^{KI}$ cells or wild-type cells overexpressing Sec23–eGFP upon transfection of mCherry or mCherry–Sec16 constructs, followed by a 20-h incubation in oleate-containing medium. Scale bars, 5 μm and 1 μm (inlay). Source numerical data and unprocessed blots are available in source data.

additional double-stranded RNAs (dsRNAs) against Rabs implicated in LD biology[36–40] and tested whether they are required for LD localization of GPAT4, fluorescently tagged at its endogenous genomic locus (Fig. 3a,b and Extended Data Fig. 2a,b). Only the depletion of Rab1, but not Rab7, Rab8, Rab18, Rab32 or Rab40,

abolished GPAT4 targeting to LDs. Expressing tagged wild-type Rab1 in cells depleted of Rab1 (with dsRNA against 5′ untranslated region) rescued GPAT4 targeting to LDs, supporting specificity of RNAi (Extended Data Fig. 3b,c). Expression of a Rab1 N124I mutant, which acts as a dominant negative by sequestering Rab1's

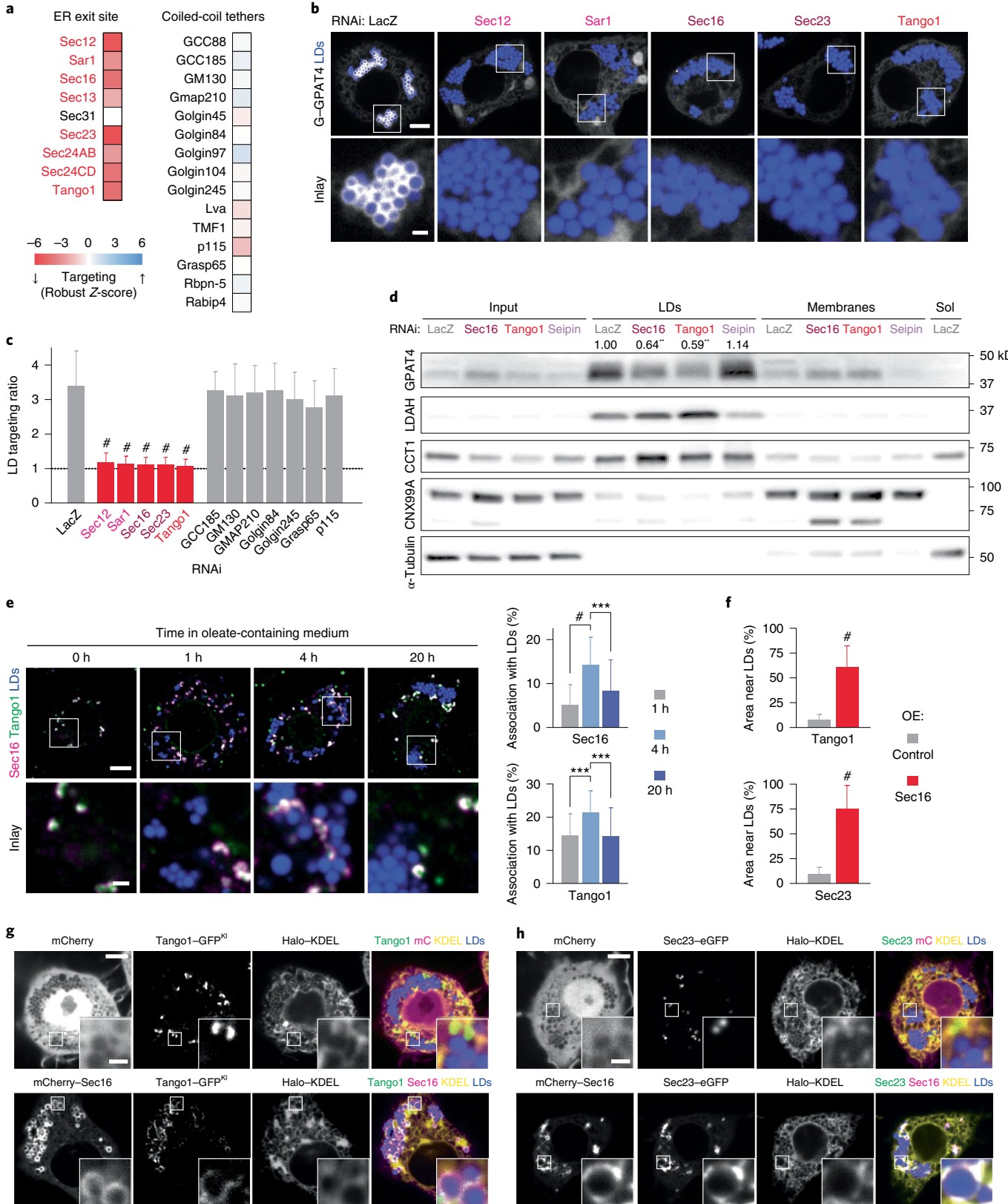

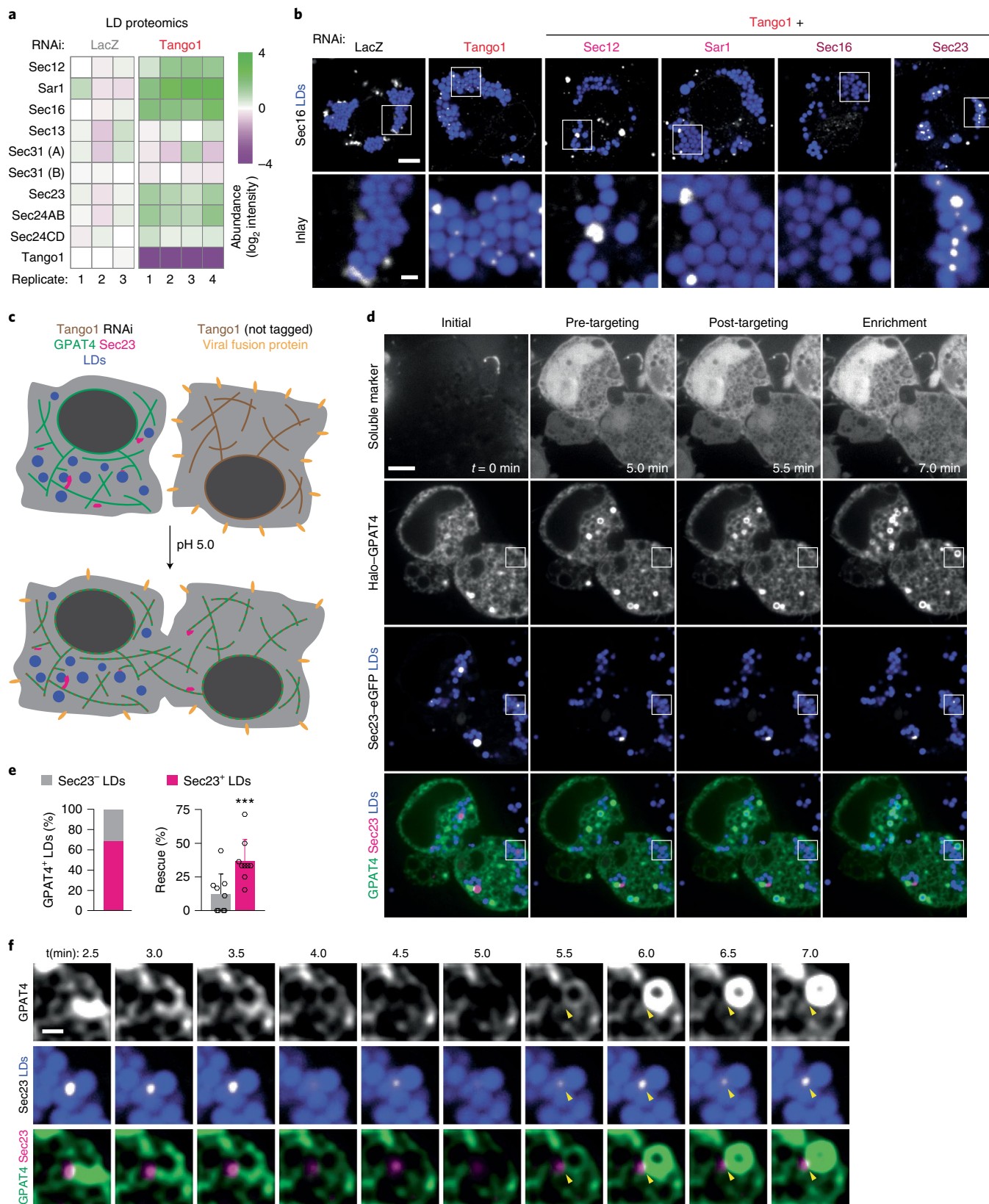

guanine nucleotide exchange factor[41], impaired endogenous GPAT4 targeting to LDs (Extended Data Fig. 4a,b).

GPAT4 targeting to LDs was reduced with depletion of Cog2 (robust $Z$-score = −3.5), Cog3 (−3.2), Cog4 (−3.3), Rint1 (−4.1), Zw10 (−2.8) and Trs20 (−4.9) membrane-tethering complex components

(Fig. 2d). In validation studies, depletion of Cog2, Cog3 and Cog4 led to much smaller LDs but did not impair LD targeting of endogenous GPAT4 (Extended Data Fig. 2a), indicating underestimation of targeting ratios in the screen. In contrast, depletion of Trs20 and Rint1 abolished LD targeting of tagged endogenous GPAT4

**Fig. 5 | GPAT4 targeting to LDs occurs via ER–LD membrane connections at Sec23-defined spots upon the rescue of Tango1 depletion. a**, ERES components enrich in LD fractions upon Tango1 depletion. Heat map for abundance of ERES organizers in LD fractions upon LacZ versus Tango1 RNAi, as measured by mass spectrometry and normalized to LacZ control. **b**, Sec16 strongly localizes around LDs upon Tango1 depletion. Immunofluorescence of Sec16 in wild-type cells upon RNAi of Tango1 or Tango1 plus another ERES component, followed by a 20-h incubation in oleate-containing medium. Representative images from three independent experiments are shown. Scale bars, 5 μm and 1 μm (inlay). **c**, Schematic diagram of cell–cell fusion assay to rescue Tango1 depletion. **d**, Representative images for the cell–cell fusion assay, showing soluble marker (mCherry) as fusion control, Halo–GPAT4 and Sec23–eGFP, at a timepoint before fusion ($t = 0$ min) as well as pre-GPAT4 targeting, post-GPAT4 targeting and enrichment phases. Scale bar, 5 μm. See also Supplementary Videos 2 and 3. **e**, Quantification of experiments in **d** 10 min after cell–cell fusion. Left: bar graph showing percentages of LDs that undergo rescue of GPAT4 targeting that are marked (or not marked) by Sec23 puncta. Right: bar graph compares percentages of Sec23-negative and Sec23-positive LDs that undergo GPAT4 targeting rescue. Mean ± s.d., $n = 9$ cells examined over seven independent experiments. Two-tailed, paired Student's $t$-test, ***$P = 0.0004$. **f**, Inlay of the imaging experiment in **d** showing Sec23 spot on LD and the apparent ER–LD membrane connection through which GPAT4 targeting rescue occurs. Scale bar, 1 μm. See also Supplementary Videos 2 and 3. Source numerical data are available in source data.

(Fig. 3a,b). Depletion of the other components of TRAPP complexes (for example, Trs23, Trs33, Bet5, Trs85, Trs120 or Trs130) or NRZ/NZZ complexes (for example, Rod, Zw10 or Zwilch) did not impair LD targeting of endogenous GPAT4 (Figure 3b and Extended Data Fig. 2a) despite efficient RNAi (Extended Data Fig. 2b). Expressing a fluorescently tagged Rint1 in cells depleted of Rint1 was sufficient to rescue GPAT4 targeting to LDs (Extended Data Fig. 3b,c). Depleting Vps52 and Vps53 (components of GARP complex) also did not affect GPAT4 targeting to LDs. Thus, the membrane-tethering factors Trs20 and Rint1 were required for GPAT4 targeting.

In vesicular fusion, four SNARE proteins (one from each of Qa, Qb, Qc and R classes) assemble to fuse membranes[42,43], and NSF and αSNAP disassemble the post-fusion SNARE complex[44–47]. In the screen, depleting several Qa SNAREs (Syx5, robust $Z$-score = −6.7; Syx13, −3.4; Syx18, −4.5) and Qb SNAREs (membrin, −2.7; Sec20, −2.9) and a single Qc (Bet1, −6.1) and R SNARE (Ykt6, −3.9) reduced GPAT4 targeting to LDs (Fig. 2d). In experiments with additional dsRNAs and endogenous GPAT4 knock-in cells (Fig. 3a,b and Extended Data Fig. 2a), depletion of candidates reduced but did not abolish LD targeting of GPAT4, except for a single SNARE of each class. Specifically, depletion of Syx5 (Qa), membrin (Qb), Bet1 (Qc) or Ykt6 (R) abolished GPAT4 targeting. Depleting NSF (robust $Z$-score = −3.3) or αSNAP (−6.4), but not NSF2, γSNAP1 or γSNAP2, reduced GPAT4 targeting to LDs (Fig. 3b and Extended Data Fig. 2a). Expressing wild-type Syx5 or Bet1 in cells depleted of these SNAREs rescued GPAT4 targeting to LDs (Extended Data Fig. 3b,c). Expressing dominant-negative Syx5 or NSF mutants (Syx5 1–445 truncation mutant missing the transmembrane segment[48] or NSF-E329Q mutant defective in ATP hydrolysis[47]) impaired LD targeting of endogenously tagged GPAT4 (Extended Data Fig. 4a,b).

Depletion of the membrane fusion machinery resulted in co-localization of endogenous GPAT4 with ER (Extended Data

Fig. 3a), indicating GPAT4 insertion into the ER. Immunoblot analysis corroborated this result, as depletion of Trs20, Rab1, Rint1, Syx5 or Bet1 significantly reduced endogenous GPAT4 amount in LD fractions without reducing total or microsomal GPAT4 (Fig. 3c and Extended Data Fig. 5a).

To test whether depletion of the fusion machinery affects GPAT4 mobility in the ER, thereby indirectly impacting its targeting to LDs, we assayed for protein dynamics with fluorescence recovery after photobleaching (FRAP). Mobility of fluorescently tagged GPAT4 in the ER was comparable in cells depleted for Rab1, Rint1 or Syx5, with or without treatment with oleic acid (Extended Data Fig. 4c–e and Supplementary Video 1).

Depletion of membrane fusion machinery Trs20, Rab1, Rint1, Syx5 or Bet1 did not affect the LD delivery of cytosolic cargoes Lsd1, CGI-58 or CCT1 (Fig. 3d,e and Extended Data Fig. 5f,g), indicating that it specifically affected ER-to-LD targeting. The targeting phenotype was specific to the late ER-to-LD targeting pathway, as depletion of these proteins impaired LD targeting of Ldsdh1 (Fig. 3d,e) and HSD17B11 (Extended Data Fig. 3b–e) but not of early cargoes LDAH and Ubxd8 (Fig. 3c–e and Extended Data Fig. 5f,g).

**Membrane-fusion factors Rab1 and Rint1 localize to LDs.** To determine if the identified membrane-fusion machinery acts directly at LDs, we analysed the localization of these factors in cells. An analysis of the published proteome of murine liver LDs[49] revealed that three of the four SNARE orthologues required for late ER-to-LD protein targeting in *Drosophila* cells (that is, Stx5 (orthologue of Syx5), Bet1l (orthologue of Bet1), and Ykt6) were enriched in LD fractions (Extended Data Fig. 6a). However, because SNAREs act transiently in numerous membrane-fusion reactions in cells, making them difficult to analyse, we focused on the localizations of Rab1, Rint1 and Trs20.

Consistent with reports of Rab1 enrichment in LD proteomes[50,51], enhanced green fluorescent protein (eGFP)–Rab1 formed

**Fig. 6 | Seipin depletion allows for late targeting proteins to target early from the ER to LDs in the absence of fusion machinery or ERES components. a**, GPAT4 targeting occurs at ER–LD connections independent of seipin. FRAP experiment of transiently expressed Halo–GPAT4 on LDs in endogenous GFP–seipin knock-in (KI) cells, after 6–10 h incubation in oleate-containing medium. Top: inlay images. Bottom: whole cell view. Yellow arrowheads indicate apparent ER–LD connections independent of seipin. Scale bars, 5 μm and 1 μm (inlay). Representative images from five independent experiments are shown. See also Supplementary Videos 4 and 5. **b**, Late targeting proteins target LDs early in the absence of seipin. Confocal imaging of live seipin knock-out (KO) cells transiently transfected with eGFP-tagged constructs at given timepoints after 1 mM oleic acid treatment. LDs were stained with MDH. Representative images are shown. Percentage of cells with LD targeting are indicated (mean, $n = 3$ independent experiments, 8–13 cells each). Scale bars, 5 μm and 1 μm (inlay). **c**, Absence of seipin provides an alternative pathway for late ER-to-LD targeting. Confocal imaging of live wild-type (WT) or seipin KO cells upon RNAi of ERES or fusion-machinery components, followed by transient transfection with eGFP-tagged constructs and a 20-h incubation in oleate-containing medium. Scale bars, 5 μm and 1 μm (inlay). **d**, Bar graph showing targeting ratios from the imaging experiment in **c**. Mean ± s.d., $n =$ (left to right) 85, 33, 41, 42, 41, 42, 42 and 40; 84, 29, 35, 46, 48, 44, 49 and 44; 89, 42, 38, 41, 46, 47, 48 and 43 cells examined over three independent experiments. One-way ANOVA with Bonferroni correction, **$P < 0.01$ (from left to right: 0.0029 and 0.0067), ***$P = 0.0005$, compared with LacZ. **e**, Bar graph showing percentages of cells with LD targeting after a 0.5-h incubation in oleate-containing medium. Mean ± s.d., $n = 6$ experiments for LacZ and 3 for the rest. One-way ANOVA with Bonferroni correction, no significant differences. Representative images are shown in Extended Data Fig. 8a. Source numerical data are available in source data.

a ring-like intensity around LDs (Fig. 3f), co-localizing with the LD protein Rab18 (ref. [52]) (co-localization coefficient $R = 0.85$). In comparison, correlations between Rab1 or Rab18 intensity with the ER marker Halo–KDEL were lower ($R = 0.58$ and $0.52$, respectively) despite close association of ER and LDs (Fig. 3f and Extended Data Fig. 6d).

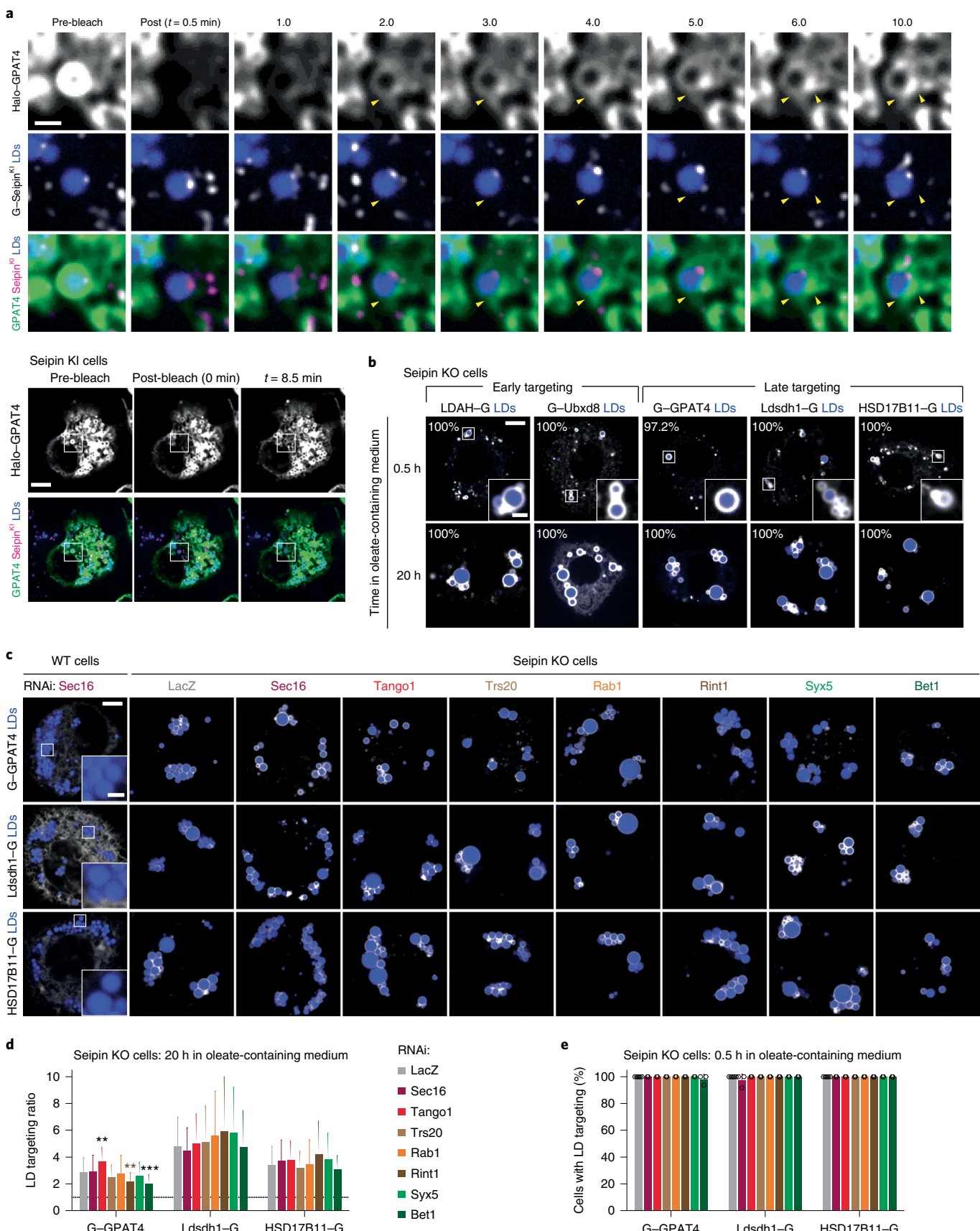

mCherry–Rint1 formed punctate intensities near LDs (Fig. 3g). Three-dimensional reconstruction (Fig. 3h and Extended Data Fig. 6b,c) suggests that Rint1 (magenta) occupies the space between Rab1 on LDs (green) and the ER (yellow). Trs20, which localized to the cytosol when expressed alone, robustly localized to LDs when co-expressed with Rint1, suggesting that Rint1 recruits Trs20 to LDs (Extended Data Fig. 6e). In contrast, Zw10, which can form a membrane-tethering complex with Rint1 in other systems[40] but was not required for GPAT4 targeting to LDs in *Drosophila* cells, localized to the cytosol (Extended Data Fig. 6f).

**ER exit sites are required for late ER-to-LD targeting.** Membrane fusion is often spatially organized to specific domains of organelles. We noted that a second category of membrane-trafficking factors required for GPAT4 targeting included genes involved in ER exit site (ERES) organization and function (Extended Data Fig. 1f). ERES are specialized ER domains that form transport carriers with protein cargoes destined for secretion via the Golgi apparatus[53]. Our screen identified most ERES proteins as required for GPAT4 targeting to LDs, including Sec12, Sec16, Tango1 and COPII coat components (Sar1, Sec23, Sec24AB, Sec24CD and Sec13) (Fig. 4a). In contrast, other proteins implicated in secretory trafficking (for example, coiled-coil tethers) were not required for GPAT4 targeting to LDs.

Depletion of ERES components in cells expressing endogenously tagged GPAT4 confirmed their requirement for GPAT4 targeting to LDs (Fig. 4b,c and Extended Data Fig. 2b). Upon depletion of Sar1, Sec16 and Tango1, endogenously tagged GPAT4 co-localized with ER (Extended Data Fig. 3a). Defective LD targeting of GPAT4 upon the depletion of Sec12, Sar1 or Sec23 was rescued by re-expressing wild-type proteins, indicating specificity of RNAi knockdowns (Extended Data Fig. 3b,c). Additionally, GPAT4 diffusion in the ER was not affected by RNAi of Sar1 or Tango1 (Extended Data Fig. 4c–e and Supplementary Video 1).

The effect of ERES component depletion on ER-to-LD targeting was specific to late cargoes (Ldsdh1 and HSD17B11; Extended Data Fig. 7a–d) and did not affect early cargoes (LDAH and Ubxd8; Extended Data Fig. 7e) or proteins targeting from the cytosol (Lsd1, CGI-58 and CCT1; Extended Data Fig. 7f). Depletion of Sec16 or Tango1 (but not LacZ or seipin controls) reduced the abundance of endogenous GPAT4, but not LDAH or CCT1, purified with LDs in subcellular fractions (Fig. 4d).

**ER exit site proteins localize to LDs.** To further test if ERES are involved in late ER-to-LD protein targeting, we used immunofluorescence to localize Sec16 and Tango1 during LD maturation. Many ERES were not localized to LDs, presumably because they operate in canonical protein export to the Golgi apparatus (Fig. 4e). However, some ERES localized to apparent contact sites of the ER and LDs. Importantly, the proportion of ERES associated with LDs increased transiently around the time of late ER-to-LD protein targeting (~4 h after oleic acid supplementation). Increased association of Sec16 and Tango1 with LDs was accompanied by increased ERES numbers per cell (Extended Data Fig. 7g). Furthermore, overexpressed,

fluorescently tagged Sec16 localized around LDs and recruited endogenous, fluorescently tagged Tango1 and transiently expressed, fluorescently tagged Sec23 to LDs (Fig. 4f–h), indicating that Sec16 may act upstream of Tango1 and Sec23 at LDs.

Dominant-negative Sar1 formed a ring-like intensity around LDs, and its expression impaired LD targeting of endogenously tagged GPAT4 (Extended Data Fig. 4a,b). Localization of overexpressed Sec16 around LDs required Sec12 and Sar1 but not Sec23 (Extended Data Fig. 7h). This contrasts with previous findings that Sec16 localization to canonical ERES is independent of Sar1 (ref. [54]) and suggests differences in the ERES organization for LD and Golgi protein targeting.

**Sec23 marks ER–LD connections mediating GPAT4 transport.** To better understand how ERES components organize around LDs, we depleted cells of the key component Tango1 and tested the association of other ERES components with LDs by mass spectrometry. Strikingly, Sec12, Sar1, Sec16, Sec23 and Sec24AB were highly enriched in LD fractions from cells lacking Tango1 than LacZ controls (Fig. 5a). Immunofluorescence microscopy showed increased Sec16 association with LDs upon Tango1 depletion (Fig. 5b, two left-most panels). Similarly, Tango1 depletion increased association of the overexpressed constitutively active Sar1 H74G mutant (defective in GTP hydrolysis) and Sec23 with a subset of LDs (Extended Data Fig. 7i). Sec16 recruitment to LDs upon Tango1 depletion required Sec12 and Sar1 but not Sec23 (Fig. 5b).

To test if the increased ERES association with LDs upon Tango1 depletion represents an intermediate to ER–LD membrane bridge formation, we performed cell–cell fusion assays that allow synchronization of GPAT4 targeting[23] (Fig. 5c). Cell fusion, mediated by viral fusion protein (vesicular stomatitis virus G) on the cell surface, supplied Tango1 from wild-type cells to cells lacking Tango1 through the inter-mixing of the cytosol and ER within minutes. As expected, soluble mCherry diffused throughout cells immediately after fusion (Fig. 5d and Supplementary Video 2). Halo–GPAT4 did not target LDs before cell–cell fusion but began to enrich rapidly at a subset of LDs after fusion. On most LDs (~69%) targeted by GPAT4, we also detected a Sec23 focus (Fig. 5d,e). In some instances, we observed reticular GPAT4 signal connecting to the LD at Sec23 puncta, indicating apparent ER–LD connections (Fig. 5f and Supplementary Video 2; additional example in Supplementary Video 3).

**Seipin restricts late ER cargoes from accessing early LDs.** What prevents late targeting proteins from accessing LDs during their biogenesis? Since seipin forms a large complex with 20–24 transmembrane domains (depending on species) resulting in a 10–15-nm ring around the budding neck of forming LDs[55–57], we hypothesized it may prevent some proteins from accessing forming LDs.

To test if late ER-to-LD protein targeting occurs independently of seipin-marked ER-LD connections, we performed FRAP of Halo–GPAT4 in *Drosophila* cells expressing GFP–seipin from its endogenous genomic locus[18]. As reported[18,58], most LDs associated with one seipin punctum (Fig. 6a and Supplementary Video 4). However, fluorescence recovery of GPAT4 occurred via many apparent

**Fig. 7 | LD proteomics reveal additional late ER-to-LD targeting protein cargoes. a**, Heat map of abundance of potential ER-to-LD targeting proteins in LD fractions upon depletion of the late protein targeting machinery components or seipin (compared with LacZ control), as measured by mass spectrometry. **b**, LPCAT, ACSL5 and DHRS7B require ERES or fusion-machinery components for LD targeting. Confocal imaging of live wild-type cells upon RNAi of ERES or fusion-machinery components, followed by transient transfection with eGFP-tagged constructs and a 20-h incubation in oleate-containing medium. LDs were stained with MDH. Scale bars, 5 μm and 1 μm (inlay). **c**, Bar graph showing LD targeting ratios from the imaging experiment in **b**. Mean ± s.d., n = (left to right; top to bottom) 48, 46, 51, 51, 48 and 48; 46, 46, 49, 54, 47 and 44; 46, 44, 43, 51, 47 and 42; 39, 43, 46, 37, 32 and 30 cells examined over three independent experiments. One-way ANOVA with Bonferroni correction, #P < 0.0001, compared with LacZ. **d**, Model of ER-to-LD protein targeting. Early cargoes can access forming LDs from the ER through the LDACs, whereas late cargoes cannot. In a process mediated by the membrane-fusion machinery, including a Rab protein, membrane tethers and SNAREs, at ERES, an ER–LD bridge forms independent of seipin, allowing LD targeting of late cargoes (such as GPAT4 and LPCAT) that are crucial for lipid metabolism and remodelling on LD surfaces. Source numerical data are available in source data.

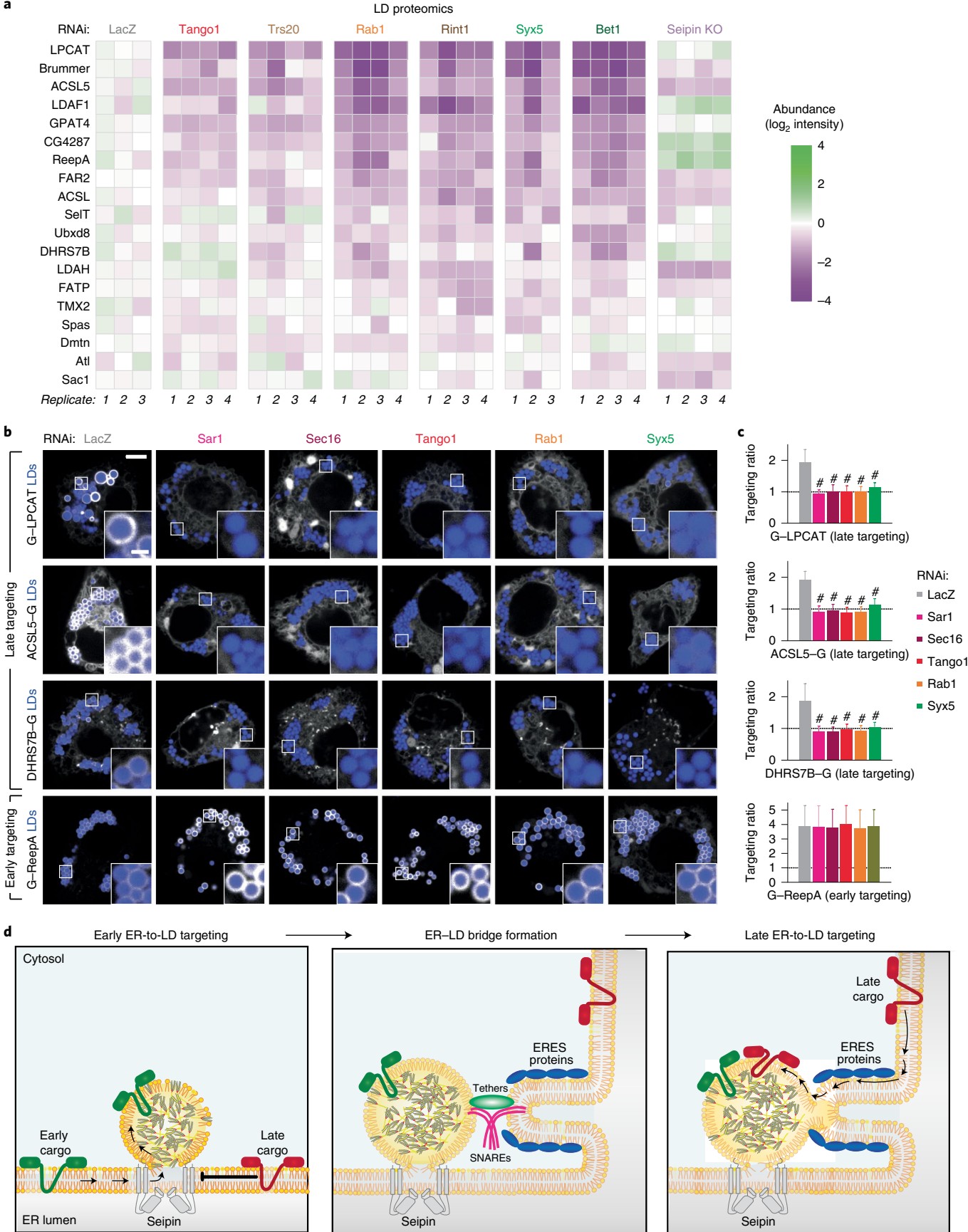

ER–LD connections not marked with seipin (arrowheads in Fig. 6a; additional example in Supplementary Video 5), suggesting GPAT4 localizes to LDs independently of the seipin-containing LDAC.

To test if seipin prevents late cargoes from accessing newly forming LDs, we measured targeting kinetics of LD cargoes in cells lacking seipin. Unlike in wild-type cells (Fig. 1), each of the analysed early (LDAH and Ubxd8) and late ER-to-LD cargoes (GPAT4, Ldsdh1 and HSD17B11) targeted LDs as early as 30 min after LD induction in seipin knock-out cells (Fig. 6b). Depletion of ERES or membrane-fusion machinery components (Sec16, Tango1, Trs20, Rab1, Rint1, Syx5 or Bet1) did not impair GPAT4 targeting to LDs in seipin knock-out cells (Fig. 6c,d), unlike in wild-type cells (Figs. 3,4). Specifically, late cargoes targeted LDs during formation in the absence of seipin when the ERES or membrane fusion machinery proteins were depleted (Fig. 6e and Extended Data Fig. 8a). Depletion of the membrane-fusion machinery did not alter the endogenous seipin foci (Extended Data Fig. 8b). Thus, seipin functions as a negative regulator of protein targeting to forming LDs, restricting the access of specific cargoes. This also indicates that depletion of late ER-to-LD protein targeting machinery does not impair the ability of the cargoes to move to LDs but instead abolishes their path to LDs.

**Systematic identification of late ER-to-LD targeting cargoes.** Identification of the machinery for late ER-to-LD protein targeting enabled us to screen for cargoes of this pathway. We individually depleted the ERES or membrane-fusion machinery (Tango1, Trs20, Rab1, Rint1, Syx5 or Bet1), isolated LDs and analysed their proteomes. Seipin knock-out cells and LacZ RNAi served as controls. Protein levels of each factor targeted by RNAi were reduced (Extended Data Fig. 8c). Focusing on LD proteins[50] with two or four consecutive predicted transmembrane domains that may form membrane-embedded hairpins, we found approximately ten proteins whose amounts in LD fractions were reduced by the depletion of late ER-to-LD protein targeting machinery components. These proteins included LPCAT, ACSL5, ReepA and DHRS7B, in addition to GPAT4 (Fig. 7a). LD localization of overexpressed, fluorescently tagged LPCAT, ACSL5 and DHRS7B were strongly impaired in cells lacking Sar1, Sec16, Tango1, Rab1 or Syx5 (Fig. 7b,c). ReepA targeting was not impaired, but it targeted LDs early instead, unlike LPCAT, ACSL5 and DHRS7B, which targeted LDs late (Extended Data Fig. 8d,e). Additional candidate proteins requiring the late ER-to-LD targeting machinery are listed in Supplementary Table 4.

## Discussion

We addressed a major gap in the understanding of protein targeting in eukaryotic cells: how proteins localize from the bilayer ER membrane to the LD monolayer. Using *Drosophila* cells, we uncovered two distinct mechanisms for ER-to-LD protein targeting: early targeting with ER proteins transiting through LDACs during LD formation, and late targeting via independently established membrane bridges that connect the ER with mature LDs. We identified the membrane fusion machinery mediating the late targeting pathway, and ERES as the site of the ER–LD bridge formation. Additionally, we identified cargoes of each targeting pathway.

Artificial ER-embedded hairpins, such as *LiveDrop*[18], HPos[16] and the LDAC component LDAF1 (ref. [19]), access LDs during their formation. We show that early ER-to-LD targeting occurs for other cellular proteins, including Ubxd8, LDAH and ReepA. How these ER proteins, but not others, access forming LDs is unclear. Current data suggest that the oligomeric seipin ring at LDACs restricts some proteins from accessing nascent LDs[55–57] while permitting others (Fig. 7d).

The LDAC barrier at LDs necessitates an alternative pathway for late ER-to-LD protein targeting. Previous studies in *Drosophila* cells suggested that GPAT4 traffics to LDs via ER–LD membrane bridges[17]. The estimated speed of GPAT4 targeting to LDs was consistent with diffusion within membranes but too fast for vesicular trafficking[23]. These findings suggested that cells can generate ER–LD membrane bridges through an unknown mechanism.

We now identified protein components required for GPAT4 targeting. Among Rab proteins, Rab1 was specifically required. Previously, Rab1 was implicated in tethering COPII-coated vesicles to the *cis*-Golgi apparatus by interacting with coil-coiled tethers, such as p115 and GM130 (refs. [59–61]). Given the localization of Rab1 on LDs[50,51], we suspect Rab1 acts as a molecular switch priming LDs for fusion with the ER. Alternatively, it may facilitate the extrusion of late cargoes at ERES[62]. We also found that the membrane-tethering complex component Rint1, which was proposed to establish ER–LD contacts[40], localizes around LDs and is required for late ER-to-LD protein targeting, whereas p115 and GM130 are dispensable.

The SNAREs necessary for late ER-to-LD targeting constitute a putative SNAREpin: Syx5 (Qa SNARE), membrin (Qb), Bet1 (Qc) and Ykt6 (R). The capacity of this combination to fuse the ER membrane with an LD has not been tested, and which of the SNAREs act at LDs is unclear. Ykt6 is a candidate since it is anchored to membranes via lipid modification[63] rather than a transmembrane domain, which would be incompatible with LD architecture. Also, Ykt6 was identified as a potential LD protein in systematic studies of LD proteome[49,51]. We and others previously showed that Arf1/COPI proteins are required for LD targeting of GPAT4 and ATGL[23–25]. Although we do not know the sequence of their action, Arf1/COPI proteins may modify LD surface to accommodate fusion factors, such as Rab1 or Ykt6.

Unexpectedly, our screen identified most ERES components, including Sec12, Sar1, Sec16, Sec23 and Tango1[64–66], as required for late ER-to-LD protein targeting. Consistent with observations that ERES localize near LDs[25], we found transient association between ERES and LDs around the time that late ER-to-LD protein targeting begins. Depletion of Tango1 resulted in accumulation of various ERES components on a subset of LDs. Since the rapid rescue of GPAT4 targeting in cell–cell fusion experiments occurred selectively at these LDs, accumulation of ERES proteins at LDs may represent intermediates of ER–LD bridge formation. Indeed, we observed reticular GPAT4 signal connecting to LDs at these sites. This may explain why ATGL co-localizes with ERES when COPI machinery is impaired[25].

An attractive unifying model based on these and other findings[66–68] is that the ER forms tubular carriers at ERES that connect to different target organelles. In the case of secretory trafficking, formation of such tubules allows for secretion of large cargoes, such as collagens or lipoproteins[66,69]. For late ER-to-LD protein targeting, tubular structures at ERES could fuse with LDs to form membrane bridges, allowing membrane-embedded proteins to traverse (Fig. 7d) and accumulate on LDs[70].

The function of the temporal segregation of ER-to-LD protein targeting into early and late targeting pathways remains speculative. Inhibition of early targeting may allow for favourable control of biophysical conditions that promote LD budding by preventing protein crowding on nascent LDs[71]. Indeed, in seipin-deficient cells, small, aberrant LDs form throughout the ER with abnormal protein content[18,58,72]. In turn, late ER-to-LD targeting may enable remodelling of mature LD composition, since many lipid-metabolizing enzymes, such as ATGL, PNPLA3 and LPCAT, appear to follow this pathway.

One limitation of our studies of ER-to-LD protein targeting is that most mechanisms were elucidated in *Drosophila* cells; additional testing for evolutionary conservation is required. Nevertheless, elements of late ER-to-LD protein targeting appear to be conserved, as Arf1/COPI proteins are required for targeting of specific proteins in both flies and humans[23–25]. Importantly, proteins involved in metabolic diseases such as HSD17B13 (ref. [7]) (an orthologue of the *Drosophila* HSD17B11) may utilize the late ER-to-LD targeting

pathway, highlighting the importance of understanding these targeting mechanisms for possible therapeutics.

## Online content

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

## Methods

**Cell lines and cell culture.** The *Drosophila* cells used in this study belong to the S2R+ cell line (sex: male) and were provided by Dr Norbert Perrimon (Harvard Medical School). Cells were cultured at 26 °C in Schneider's *Drosophila* medium (Gibco, #21720001) supplemented with 10% foetal bovine serum (Gibco), 25 units ml⁻¹ penicillin and 25 µg ml⁻¹ streptomycin. Cells were maintained by splitting 1:6–1:12 every 3–4 days.

**Special reagents.** Janelia Fluor (JF) 646 and 549 HaloTag Ligands were gifts from Dr Luke Lavis (Janelia Research Campus, USA). Anti-*dm*LDAH used for western blot experiments[28] was a gift from Dr Mathias Beller (Heinrich Heine Universität Düsseldorf, Germany). Anti-*dm*Sec16 used for immunofluorescence experiments[54] was a gift from Dr Catherine Rabouille (Hubrecht Institute, the Netherlands). Anti-*dm*Tango1 used for immunofluorescence experiments[73] was a gift from Dr Sally Horne-Badovinac (University of Chicago, USA).

Oleic acid (10 mM; OA solution) was prepared by dissolving 1.98 g of essentially fatty-acid-free BSA in 10 ml PBS, adding 31.74 µl of oleic acid drop by drop and shaking at 37 °C for 1 h. Solution was sterile filtered (0.22 µm) before use. All oleic acid treatments were performed with 1 mM final concentration.

**Genome-scale RNAi imaging screen.** *Drosophila* S2R+ cells stably overexpressing eGFP–GPAT4 were subjected to a genome-scale library of dsRNA in imaging-compatible 384-well plates (PerkinElmer, #6057300) two times, prepared by the HMS Drosophila RNAi Screening Center (DRSC 2.0 genome-wide screening library). The library targets approximately 13,900 genes approximately one to two times and consists of 66 384-well plates with 250 ng of dsRNA in 5 µl per well. Confluent cells were resuspended to 60 × 10⁴ cells ml⁻¹ in Schneider's *Drosophila* Medium (Gibco, #21720001) without serum supplementation. Ten microlitres of the cell suspension was dispensed into the dsRNA plates using Thermo Scientific Matrix WellMate Microplate Dispenser. After mixing the contents gently, plates were sealed with parafilm and placed in a 'wet chamber' (airtight container with wet paper towels) in a 26 °C incubator for 50 min. Then, 30 µl of Schneider's *Drosophila* Medium supplemented with 10% foetal bovine serum, 100 units ml⁻¹ of penicillin and 100 µg ml⁻¹ of streptomycin was added to each well, and plates were sealed with parafilm and placed in the wet chamber for 3.75 days.

After RNAi, 6 µl of 10 mM OA solution and 14 µl of fresh medium were dispensed to each well. After 20 h, wells were washed once with 50 µl of PBS. Of note, for each aspiration, about half the liquid (~50 µl) was left to avoid disrupting cells. Cells were fixed for 25 min with 50 µl of freshly prepared 8% paraformaldehyde in PBS solution (final concentration of 4%) at room temperature and then washed with 70 µl of PBS three times. Seventeen microlitres of 1 µM SiR-DNA nuclear stain (Spirochrome, #SC007) and 133 µM monodansylpentane LD stain (AUTOdot; Abcepta, #SM1000b) in PBS were added to each well (final concentration 0.25 µM SiR-DNA & 33.3 µM AUTOdot) and incubated for 35 min. Finally, each well was washed with 70 µl of PBS three times, and 25 µl of PBS was added (final volume ~75 µl) for imaging.

For automated confocal imaging, we used the GE IN Cell Analyzer 6000 Cell Imaging System with robotics support for automated plate loading. Using the IN Cell Analyzer software, three channel images (FITC for eGFP–GPAT4, Cy5 for nuclei and DAPI for LDs) were taken in eight fields per well at the manually determined offset from auto-focusing for each plate using 60× objective.

**Plasmids.** PCR of the insert was performed using PfuUltra II Fusion Hotstart DNA Polymerase (Agilent Technologies, #600672), following the manufacturer's protocol. Purified PCR product was cloned into an entry vector using the pENTR/D-TOPO Cloning Kit (Invitrogen, #K240020) and subsequently into a destination vector from the *Drosophila* Gateway vector collection system (Murphy laboratory, Carnegie Mellon University) using the Gateway LR clonase Enzyme mix (Invitrogen, #11791019).

Mutagenesis was performed using the QuikChange II XL Site-Directed Mutagenesis Kit (Agilent, #200521) in an entry vector, which was then cloned into a destination vector for expression.

All final plasmids were verified by restriction analysis and sequencing of the insert. PCR template and primer sequences are provided in Supplementary Table 5.

**Transfection.** Cells were transfected with Effectene Transfection Reagent (Qiagen, #301425), following the manufacturer's protocol. When co-transfecting with more than one plasmid, equal amount (in µg) of the plasmids was used. Any further treatments took place 26 h after transfection.

**In vitro dsRNA synthesis.** Genomic DNA was isolated using the DNeasy Blood & Tissue Kit (Qiagen, #69504). PCR was performed using primers containing the T7 promoter sequence (on both forward and reverse primers) with PfuUltra II Fusion Hotstart DNA Polymerase (Agilent Technologies, #600672). PCR products with the expected size were separated on a 1% agarose gel, and the MEGAscript T7 Transcription Kit (Invitrogen, #AM1334) was used for in vitro transcription. RNA was purified using the RNeasy Mini Kit (Qiagen, #74104). Primer sequences are provided in Supplementary Table 6.

**RNAi.** Cells were spun down at 300*g* for 5 min and resuspended with Schneider's *Drosophila* Medium (Gibco, #21720001) without serum supplementation at 60 × 10⁴ cells ml⁻¹. dsRNA was added at 20 ng µl⁻¹ to the plated cells. After carefully mixing the contents, plates were sealed with parafilm and placed in a wet chamber inside a 26 °C incubator. After 50 min, serum-supplemented medium with three volumes of initial cell suspension was added and incubated in the wet chamber for 3.5–4 days before further treatments. For transfection, cells were transferred onto a new plate before following the transfection protocol.

**qPCR.** Total RNA from *Drosophila* S2R+ cells was isolated using the QIAGEN RNeasy Kit, according to the manufacturer's instructions, along with QIAshredder and on-column genomic DNA digestion using RNase-free DNase Set (QIAGEN #79256). Complementary DNA was synthesized using iScript cDNA Synthesis Kit (Bio-Rad #1708840), and qPCR was performed in duplicate using SYBR Green PCR Master Mix Kit (Applied Biosystems #4368706). Primer sequences are provided in Supplementary Table 6.

**Generation of cell lines.** A stable cell line overexpressing eGFP–GPAT4 was created by transfecting cells with pActin–eGFP–GPAT4–T2A–Puro^R and selecting with 10 µg ml⁻¹ puromycin for 3 days twice before cell sorting. Information on PCR template and primers used for cloning the construct is provided in Supplementary Table 5. For cell sorting, cells were suspended in sterile PBS supplemented with 1% foetal bovine serum. Using FACSAria-561 with 100-µm gating, eGFP⁺ cells (488-nm laser) were sorted into a 96-well plate (100 cells per well) containing conditioned medium (medium collected from cells growing at exponential phase, combined with equal volume of fresh Schneider's medium supplemented with 20% foetal bovine serum). After 2 weeks, cells were expanded and subjected to microscopy and western blot for verification of the cell line.

Knock-out and knock-in cell lines were created using CRISPR–Cas9, following protocols by Housden et al.[74,75]. Guide RNA sequence and PCR primer sequences for donor construct cloning are provided in Supplementary Table 5. At 1 week after transfection, cells were sorted as above. After 3 weeks, viable single-cell colonies were subjected to microscopy, western blot and genomic DNA sequencing for verification.

**Immunofluorescence.** Cells were plated on 96-square-well clear-bottom plates (Perkin Elmer) at ~50% confluency with 1 mM OA treatment. At a given OA timepoint, wells were washed twice with PBS (each wash was performed with ~100 µl liquid remaining) and fixed with 4% paraformaldehyde (Polysciences 18814-10) in for 10 min, followed by washing with 200 µl of PBS four times. After permeabilizing with 0.15% Triton X-100 and 0.15% BSA in PBS for 3 min, wells were washed four times with PBS and blocked with 7.5% normal goat serum (Cell Signaling 5425 S) in PBS for 1 h. After aspirating and leaving ~100 µl volume, 50 µl of antibody solution for the final concentration of 1:1,500 rabbit anti-*dm*Sec16 (ref. [54]) or guinea pig anti-*dm*Tango1 (ref. [73]) in 5% normal goat serum in PBS was added and incubated for 1.5 h. Wells were then washed four times with 0.2% BSA in PBS solution and once with 200 µl of 5% normal goat serum in PBS. We then added 50 µl of secondary antibody solution for a final concentration of 1:1,000 (Alexa Fluor 488 goat anti-guinea pig IgG, Thermo Scientific A-11073; Alexa Fluor 647 goat anti-rabbit IgG, Thermo Scientific A-21244) in 5% normal goat serum in PBS and incubated for 1.5 h. Wells were then washed four times with 0.2% BSA in PBS and four times with PBS. Finally, 0.10 µl of AutoDot (Abcepta SM1000b) in 200 µl of PBS was added for LD staining before imaging.

**Cell–cell fusion assays.** Wild-type *Drosophila* S2R+ cells were transfected with constructs encoding VSV G (viral fusion protein)[23] and mCherry (soluble marker) and mixed 1:1 with cells that underwent RNAi of Tango1 for 3.5 days, followed by transfection with constructs encoding Halo–GPAT4 and Sec23–eGFP for 1.5 days and incubation in 1 mM oleate-containing medium for 8 h. Cells were prepared for imaging in 96-square-well clear-bottom plates (Perkin Elmer) as above. After taking pre-fusion images, fusion was initiated by removing medium and adding pH 5.0 buffer (10 mM Na₂HPO₄, 10 mM NaH₂PO₄, 150 mM NaCl, 10 mM MES and 10 mM HEPES) for 40 s on the microscope stage. After aspirating out the low-pH buffer, regular growth medium was added to cells for timelapse imaging.

**Fluorescence microscopy.** Cells that have undergone transfection or RNAi were resuspended and combined with an equal volume of fresh medium onto a 35-mm dish with 14-mm No. 1.5 coverslip bottom (MatTek Life Sciences, #P35G-1.5-14-C), coated manually with 0.1 mg ml⁻¹ Concanavalin A. Cells were allowed to settle for 1 h at 26 °C before further treatments, such as with 1 mM OA. Unless otherwise indicated, cells were imaged 20 h after OA treatment. LDs were stained with 100 µM monodansylpentane (AUTOdot; Abcepta, #SM1000b) unless otherwise noted, for instance with 1:1,000 HCS LipidTOX Deep Red Neutral Lipid Stain (Thermo Fisher Scientific, H34477), 10 min before imaging. For Halo constructs, cells were incubated with JF dyes 1 h before imaging and washed once with PBS.

Nikon Eclipse Ti inverted microscope, featuring CSU-X1 spinning disk confocal (Yokogama) and Zyla 4.2 PLUS scientific complementary metal-oxide semiconductor (Andor), was used for spinning disk confocal microscopy.

NIS-elements software (Nikon) was used for acquisition control. Plan Apochromat VC 100× oil objective (Nikon) with 1.40 NA was used, resulting in 0.065-μm pixel size. Solid-state excitation lasers (405 nm, blue, Andor; 488 nm, green, Andor; 561 nm, red, Cobolt; 637 nm, far red, Coherent) shared a quad-pass dichroic beam splitter (Di01-T405/488/568/647, Semrock), whereas emission filters were FF01-452/45, FF03-525/50, FF01-607/36 and FF02-685/40 (Semrock), respectively.

For FRAP experiments, Bruker Mini-scanner module was used. To photobleach eGFP–GPAT4 in the ER, 488-nm laser was applied at 20% power for 300 μs to a 3 nm-by-3 nm square area. To photobleach Halo–GPAT4 conjugated to JF549 on LDs, 561-nm laser was applied at 20% power for 200 μs to a 2-nm-diameter circular area.

**Fractionation of cells.** Cells were washed once with PBS at room temperature, and all subsequent steps were performed on ice and with buffer chilled to 4 °C. Cell pellets were suspended in 1 ml of 250 mM sucrose buffer containing 200 mM Tris–HCl (pH 7.4), 1 mM MgCl$_2$ (pH 7.4) and cOmplete Mini EDTA–protease inhibitor cocktail (Roche, #4693159001) and passed through 25 G syringe 30 times. Then, 1 unit μl$^{-1}$ of benzonase nuclease (Millipore, #E1014) was added for 10 min. Five percent of the total volume was saved as whole-cell lysate ('input'). For the rest, unbroken cells and nuclei were removed by centrifuging for 5 min at 1,000g at 4 °C. Top lipid layer and the supernatant were moved to a 5-ml tube (Open-Top Thinwall Ultra-Clear Tube, 13 × 51 mm, Beckman Coulter, #344057), and an additional 1.5 ml of the 250 mM sucrose buffer was added. Then, 2.5 ml of 50 mM sucrose containing 200 mM Tris–HCl (pH 7.4), 1 mM MgCl$_2$ and cOmplete Mini EDTA–protease inhibitor cocktail was layered on top. The two-step sucrose gradient was centrifuged for 16–20 h at 100,000g at 4 °C using the SW 55 Ti Swinging Bucket rotor (Beckman Coulter, 342194).

Top of the tube (~5 mm; 500 μl) was sliced using a Beckman Coulter tube slicer, the content of which was taken as 'LD fraction'. Supernatant was taken as 'soluble fraction', and the pellet resuspended in 500 μl of 250 mM sucrose buffer was taken as 'membrane fraction'.

For further analysis with immunoblotting or mass spectrometry, proteins from the fractions were precipitated. One millilitre of methanol and 250 μl of chloroform were sequentially added to ~500–750 μl of a fraction with vigorous mixing after every addition. After centrifuging for 10 min at 14,000g at 4 °C, the top layer was aspirated, and 1.7 ml of methanol was added and vigorously mixed. Protein precipitation was then isolated by centrifuging for 15 min at 18,000g at 4 °C and after drying for 5 min at room temperature, resuspended in 100–250 μl of 1.5% SDS and 50 mM Tris–HCl (pH 7.4) buffer.

**Immunoblotting.** Protein concentrations were measured using the Pierce BCA Protein Assay Kit (Thermo scientific, #23225), and the amounts indicated in respective figure legends were resuspended in 1× Laemmli buffer (2% SDS, 10% glycerol, 50 mM Tris–HCl (pH 6.8), β-mercaptoethanol and 0.02% bromophenol-blue). After running samples in 4–15% gradient polyacrylamide gel (Bio-Rad, #4561084) at 100 V for 90 min in 1× Tris/glycine/SDS buffer (Bio-Rad, #161-0772), proteins were transferred to a 0.2-μm pore-size nitrocellulose membrane (Bio-Rad, #1620112) in 1× Tris/glycine buffer (Bio-Rad, #161-0771) at 70 V for 90 min in a cold room (4 °C). Membranes were blocked by incubating in 5% non-fat dry milk (Santa Cruz Biotechnology, #sc-2325) in TBS-T buffer (20 mM Tris, pH 7.6, 150 mM NaCl and 0.1% Tween-20) for 30 min at room temperature. Membranes were then incubated with 5% milk solution containing primary antibody (dilutions are indicated below) overnight in the cold room.

On the following day, membranes were washed three times with TBS-T for 10 min each, incubated with 1:5,000 secondary antibodies conjugated to horseradish peroxidase (mouse anti-rabbit IgG-HRP (Santa Cruz Biotechnology, Cat# sc-2357), mouse anti-IgG kappa binding protein-HRP (Santa Cruz Biotechnology, Cat# sc-516102), goat anti-rat IgG H&L-HRP (Abcam, Cat# ab97057)), in 5% milk solution for 1 h, and washed three times with TBS-T for 10 min each at room temperature. SuperSignal West Pico PLUS Chemiluminescent Substrate (Thermo Scientific, #34580) was applied to the membrane, and the blot was imaged using the Biorad Gel Doc XR system.

For stripping the membrane of antibodies, membrane was washed with distilled water five times for 5 min each and incubated with 100 mM citric acid solution in distilled water for 10 min at room temperature. The membrane was then re-blocked with 5% milk solution for 30 min before proceeding.

Primary antibodies and their dilutions: rabbit anti-*dm*GPAT4 (ref. [17]) (1:1,000), rabbit anti-*dm*CCT1 (ref. [76]) (1:1,000), mouse anti-*dm*CNX99A[77] (1:500; DSHB, #Cnx99A 6-2-1), mouse anti-α-tubulin (1:2,000; Sigma, T5168) and anti-*dm*LDAH[28] (1:2,000).

**Mass spectrometry.** Proteins pellets from LD-enriched fractions were resuspended in 0.1 M NaOH (Sigma-Aldrich) and subsequently neutralized using 200 mM HEPES. Solubilized proteins were reduced using 5 mM dithiothreitol (Sigma-Aldrich), pH 7.5, at 37 °C for 1 h. Reduced disulfide bonds of cysteine residues were alkylated using 15 mM iodoacetamide (Sigma-Aldrich) for 1 h in the dark. Excessive iodoacetamide was quenched using 10 mM dithiothreitol. The alkylated protein mixture was diluted six-fold (v/v) using 20 mM HEPES, pH 7.5, and digested for 16 h at 37 °C with sequencing-grade trypsin (Worthington

Biochemical) in a 1:100 trypsin-to-protein ratio. Digested peptides were de-salted using self-packed C18 STAGE tips (3 M Empore)[78]. De-salted peptides were dissolved in 0.1% (v/v) formic acid and injected onto an Easy-nLC 1000 (Thermo Fisher Scientific), coupled to an Orbitrap Exploris 480 (Thermo Fisher Scientific). Peptide separation was performed on a 500-mm self-packed analytical column using PicoTip emitter (New Objective) containing Reprosil Gold 120 C-18, 1.9-μm particle-size resin (Dr. Maisch). Chromatography separation was carried out using increasing organic proportion of acetonitrile (5–40 % (v/v)) containing 0.1 % (v/v) formic acid over a 120 min gradient at a flow rate of 300 nl min$^{-1}$.

**Statistics and reproducibility.** All statistical analysis was performed using GraphPad Prism 8. Information about sample size and type of significance test is provided in the legends. No statistical method was used to pre-determine sample size. Outliers were identified using the ROUT method at $Q = 1\%$ and excluded from further analysis

The design of the RNAi assay plates for the genome-wide screen was randomized by the Drosophila RNAi Screening Center at Harvard Medical School, and the gene targets were not cross-referenced until all automatized analysis was completed. For all other experiments, The experiments were not randomized, and the investigators were not blinded to allocation during experiments and outcome assessment.

**Analysis of genome-scale imaging screen.** A MATLAB analysis pipeline was built to analyse screen images. Nucleus, cell and LD compartments were segmented using supervised machine-learning methods, random forest pixel classifiers (http://github.com/HMS-IDAC/PixelClassifier). Three different models were trained, one for each compartment, using separate sets of annotated images (seven images per model; nuclei and cells from the Cy5 channel and LDs from the DAPI channel). Nucleus mask was used for segmenting cells as markers in a watershed algorithm. LD objects were then associated with cell objects, depending on the area of intersection. Finally, the signal in the FITC channel was corrected for auto-fluorescence by subtracting the mean value of control images and quantified inside and outside the LD mask in each segmented cell. From these measurements, we calculated LD targeting ratio for each segmented cell, defined as the ratio of the mean intensity of eGFP–GPAT4 signal inside the LD mask to that of eGFP–GPAT4 signal outside LD mask within the cell mask. Median LD targeting ratios from all segmented cells from the eight fields of the same well were determined and employed as the final readout for the corresponding dsRNA.

Robust $Z$-scores for median LD targeting ratios ($X$) were calculated using the formula below. In our screen, median was 2.147287 and median absolute deviation was 0.113917.

$$\text{Robust } Z\text{score} = \frac{X_i - \text{median}(X)}{1.4826 \times \text{median absolute deviation}(X)}$$

where median absolute deviation $= \text{median}(|X_i - \text{median}(X)|)$

**Quantification of fluorescence images.** Confocal images were quantified using FIJI software[79] to calculate LD targeting ratios. Cell boundaries were manually drawn on the basis of the fluorescence from protein channels, such as eGFP–GPAT4 (mask 1), whereas LD regions were segmented by applying an automatic threshold (Otsu method) to the LD stain channel within mask 1, followed by dilation with one pixel to include LD surfaces (mask 2). LD targeting ratios were calculated by dividing the mean intensity of the fluorescent protein channel image in mask 2 divided by that in (mask 1 – mask 2). For CCT1, nuclear boundary was manually drawn and excluded from mask for subsequent analysis. To calculate percentage of Tango1 or Sec23 area near LDs, Tango1 or Sec23 and LD channels were subjected to automatic thresholding (Otsu and Huang methods, respectively), and the ratio between the area of Tango1 or Sec23 mask overlapping with LD mask dilated by one pixel and the total area of Tango1 or Sec23 mask was calculated.

FRAP analysis was performed using the ImageJ plugins from Jay Unruh at Stowers Institute for Medical Research (Kansas City, MO). Pearson's correlation coefficient and cytofluorogram for co-localization analysis were obtained using the JACoP ImageJ plugin[80].

**Three-dimensional reconstruction of fluorescence images.** Images were taken at 0.3-μm z-stacks. For protein channels, images were de-convolved using cudaDecon (https://github.com/scopetools/cudaDecon) with corresponding point spread functions for each wavelength, and the Imaris surface tool was used for segmentation. Local background subtraction was used to retain the detailed features of the segmentation, and the diameter of largest sphere that fits into the object set depended on the laser wavelength (488 nm, 561 nm or 640 nm). For LD channel, the Imaris cells tool was used to define their number and size, using estimated diameter of 0.4 μm, background subtraction and different vesicle sizes (Region Growing). Three-dimensional Gaussian blur image processing was applied to Halo–KDEL channel to find the whole-cell edge.

**Spatial association between ERES and LDs.** The ERES–LD association was determined using the DiAna ImageJ plugin[81]. The images were pre-processed

by background subtraction and median filter before the segmentation. The fluorescence signal of ERES punctate (Sec16 or Tango1) and LDs were segmented by automatic intensity thresholding. Then, the ERES and LDs were identified as three-dimensional objects, and the number of ERES and the closest distance between the ERES and LD boundaries were determined. To exclude the interference from cell debris and non-specific labelling, only the objects with pixel size >30 pixels were considered in the analysis. Zero closest distance between ERES and LDs indicated overlapping boundaries between the two. The ratio of the ERES associated with LD was calculated by dividing the number of ERES with zero closest distance with LDs by the total number of ERES within the cell.

**Quantification of immunoblots.** Using FIJI software[79], a rectangular region of interest (ROI) was drawn around the band of interest in the control lane (LacZ RNAi), and the total intensity within the ROI was measured. ROI was sequentially moved to other lanes for measurement. After performing background subtraction, the measured intensities were normalized to the level in the control lane.

**Analysis of mass spectrometry data.** The mass spectrometry analyser operated in data-dependent acquisition mode with a top ten method at a mass-over-charge (m/z) range of 300–2,000 Da. Mass spectrometry data were analysed by MaxQuant software version 1.5.2.8 (ref. [82]) using the following setting: oxidized methionine residues and protein N-terminal acetylation as variable modification, cysteine carbamidomethylation as fixed modification, first search peptide tolerance 20 ppm, and main search peptide tolerance 4.5 ppm. Protease specificity was set to trypsin with up to two missed cleavages allowed. Only peptides longer than six amino acids were analysed, and the minimal ratio count to quantify a protein was 2. The false discovery rate was set to 5% for peptide and protein identifications. Database searches were performed using the Andromeda search engine integrated into the MaxQuant software[83] against the UniProt *Drosophila melanogaster* database containing 20,981 entries (December 2018). 'Matching between runs' algorithm with a time window of 0.7 min was employed to transfer identifications between samples processed using the same nanospray conditions. Protein tables were filtered to eliminate identifications from the reverse database and common contaminants.

To identify proteins regulated by different genotypes, the MaxQuant output files were exported to Perseus 1.5.1.6 (ref. [84]). Known contaminant and decoy sequences were removed. Projection and clustering of the dataset was performed using principal component analysis to identify potential sample outlier. The cut-off of potential principal components was set at Benjamini–Hochberg false discovery rate 5%. After removing poorly clustering replicates in the principal component analysis, intensity values were normalized to the sum of all intensities within each sample. Clustergram analysis was performed using the ComplexHeatmap clustering method in R (ref. [85]).

**Materials availability.** All unique/stable reagents used in this study are available from the lead contact with a completed Materials Transfer Agreement in accordance with the Harvard T.H. Chan School of Public Health policies.

**Reporting summary.** Further information on research design is available in the Nature Research Reporting Summary linked to this article.

## Data availability

Original screen images and quantification results are available at the Lipid Droplet Knowledge Portal[33] (http://lipiddroplet.org/). Please select 'Fly gene' under 'Query a gene' and search by gene name. The mass spectrometry proteomics data have been deposited to the ProteomeXchange Consortium via the PRIDE[86] partner repository with the dataset identifier PXD027283. All other data supporting the findings of this study are available from the corresponding author on reasonable request. Source data are provided with this paper.

## Code availability

Original code (used for quantification of screen images) has been deposited at Harvard Dataverse and is publicly available (https://doi.org/10.7910/DVN/NKJDWS).

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

## Acknowledgements

We thank the members of Farese & Walther laboratory, T. Mitchison (HMS), N. Perrimon (HMS, HHMI) and A. Salic (HMS) for helpful discussions. We also thank M. Beller (Heinrich Heine Universität Düsseldorf, Germany), S. Horne-Badovinac (University of Chicago), L. Lavis (HHMI), C. Rabouille (Hubrecht Institute, the Netherlands) and N. Perrimon for reagents; R. Tao, J. Zirin, Y. Hu, D. Yang-Zhou, S. Knight and G. Amador (DRSC, Department of Genetics, HMS) for technical support with screening; the Nikon Imaging Center (HMS) for imaging support; the Flow Cytometry Core (Department of Immunology, HMS) for the support with cell sorting; and G. Howard for editorial assistance. This work was supported by grants from the National Institutes of Health R01GM097194 (T.C.W.), 5TL1TR001101 (J.S.) and T32GM007753 (J.S.). J.S. received support from the American Heart Association Predoctoral Fellowship and the Aramont Fund for Emerging Science Research. T.C.W. is a Howard Hughes Medical Institute investigator. A.M. is a Helen Hay Whitney Foundation Postdoctoral Fellow. The DRSC-BTRR is supported by NIGMS P41 GM132087.

## Author contributions

J.S., R.V.F. and T.C.W. conceived the project. J.S., R.V.F. and T.C.W. designed the screen, and S.E.M. and M.C. provided additional input. M.C. and J.S. built the pipeline for screen image analysis. J.S. performed and analysed the screen. J.S. performed and analysed most of the experiments, and A.M. and C.-W.L. performed and analysed additional experiments. Z.W.L. performed mass spectrometry analyses. W.-C.T. performed three-dimensional reconstruction of images. C.-H.L. performed spatial analysis of LDs and ERES. J.S., R.V.F. and T.C.W. wrote the manuscript. All authors discussed the results and contributed to the manuscript.

## Competing interests

The authors declare no competing interests.

## Additional information

**Extended data** is available for this paper at https://doi.org/10.1038/s41556-022-00974-0.

**Correspondence and requests for materials** should be addressed to Robert V. Farese or Tobias C. Walther.

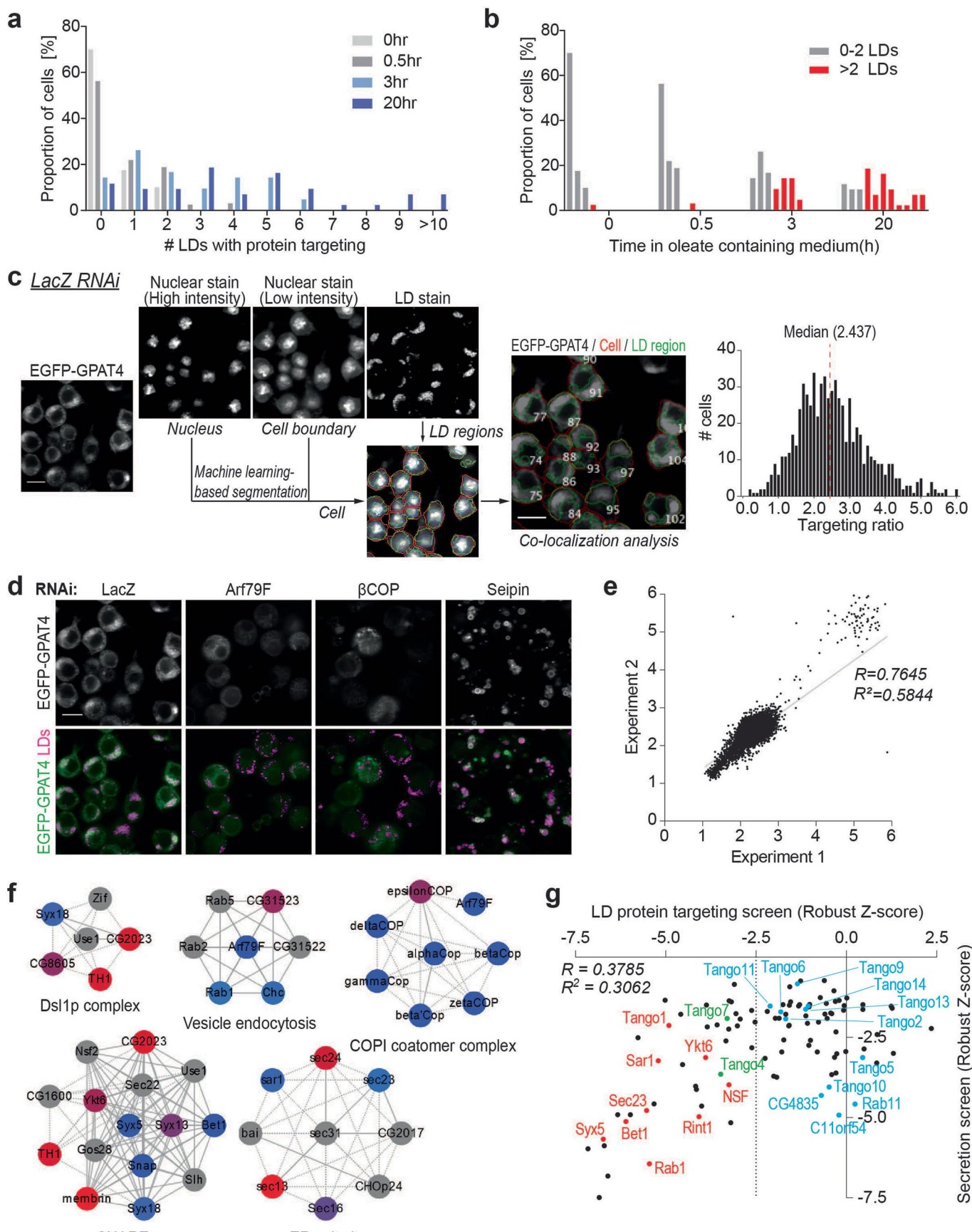

**Extended Data Fig. 1 | See next page for caption.**

**Extended Data Fig. 1 | HSD17B11 targeting to LDs and an imaging screen to identify factors required for GPAT4 targeting to LDs. a, b**, Bar graph showing percentage of cells for a given number of LDs with HSD17B11-EGFP targeting over time after 1 mM oleic acid treatment. Representative images in Fig. 1b. **c**, Schematic diagram for LD targeting ratio calculation in the imaging screen. Sample images for LacZ RNAi are shown. Machine learning is used to segment individual cells and regions of LDs from the nuclear and LD stains, which are superimposed onto EGFP-GPAT4 channel to calculate LD targeting ratio for each cell. Median value from all segmented cells in eight different fields is reported as the final readout. Scale bars, 10 μm. **d**, Representative images for screen controls. LacZ RNAi has no effect on GPAT4 targeting to LDs, whereas Arf79F and βCOP RNAi reduce and Seipin RNAi increase GPAT4 targeting to LDs. n = 528 for LacZ RNAi; n = 132 for βCOP, Arf79F, and Seipin RNAi. Scale bar, 10 μm. **e**, Genome-scale screen is reproducible. Scatter plot showing targeting ratios from two independent genome-scale experiments. Grey: linear regression. **f**, Protein complexes enriched among hits required for GPAT4 targeting to LDs (robust Z-score < −2.5) using COMPLEAT[37]. Blue to red nodes: lowest to highest robust Z-scores; grey node (non-hits, robust Z-score > −2.5). Solid line: known interaction in *Drosophila*; dashed line: known interaction in other species. Permutation test as compared to 1000 random complexes of the same size, p-value < 0.01 for all complexes shown. **g**, Comparison of robust Z-scores between the LD protein targeting screen and the secretion screen[38]. In the secretion screen, the effect of genome-scale dsRNA library on the HRP secretion of *Drosophila* S2 cells was measured using chemiluminescence. Dotted line at robust Z-score < −2.5 for LD protein targeting screen hits. Red: select genes that are hits in both screens and characterized further in this study; green: other genes that are hits in both screens; blue: secretion screen hits that are not LD targeting screen hits. Source numerical data are available in source data.

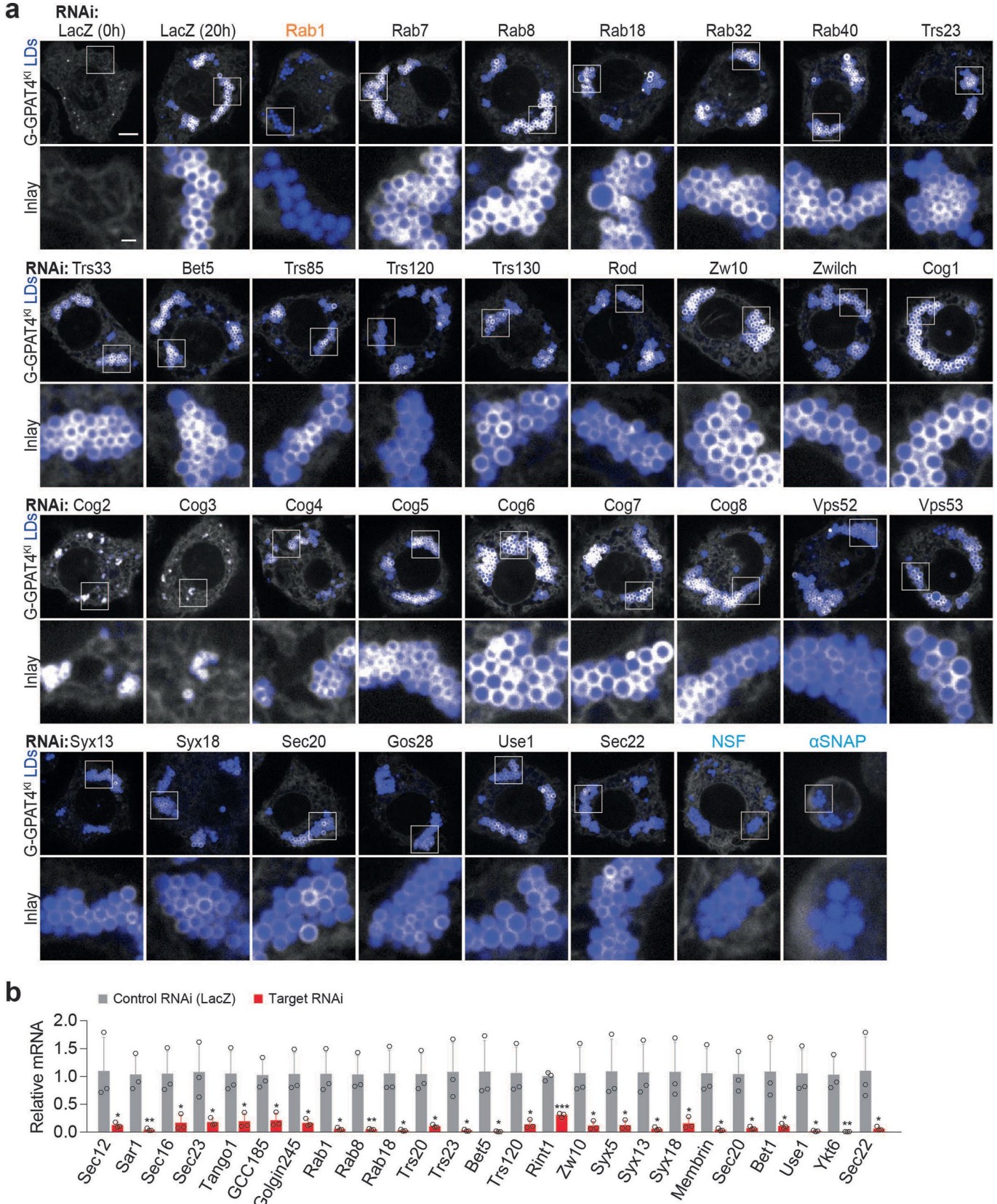

**Extended Data Fig. 2 | See next page for caption.**

**Extended Data Fig. 2 | Membrane-fusion regulator, tether, and SNAREs are required for GPAT4 targeting to LDs. a**, Depletion of specific Rab, tethering-complex components, and SNAREs abolishes GPAT4 targeting to LDs. Confocal imaging of live EGFP-GPAT4 endogenous knock-in cells upon RNAi of membrane fusion machinery components, 20 h after 1 mM oleic acid treatment (except for 0 h timepoint for LacZ RNAi). LDs were stained with MDH. Representative images are shown. Scale bar, 5 and 1 μm (inlay). Quantification of targeting ratios is shown in Fig. 3b. **b**, Quantitative PCR to verify RNAi. Mean ± SD, n = 3. Two-tailed Student's t-test for each gene, *p < 0.05 (from left to right: 0.0497, 0.0241, 0.0369, 0.0258, 0.0104, 0.0180, 0.0125, 0.0140, 0.0122, 0.0264, 0.0294, 0.0271, 0.0250, 0.0462, 0.0259, 0.0430, 0.0172, 0.0104, 0.0369, 0.0142, 0.0420), **p < 0.01 (from left to right: 0.0070, 0.0076, 0.0051), ***p = 0.0001, compared to Control (LacZ) RNAi. Source numerical data are available in source data.

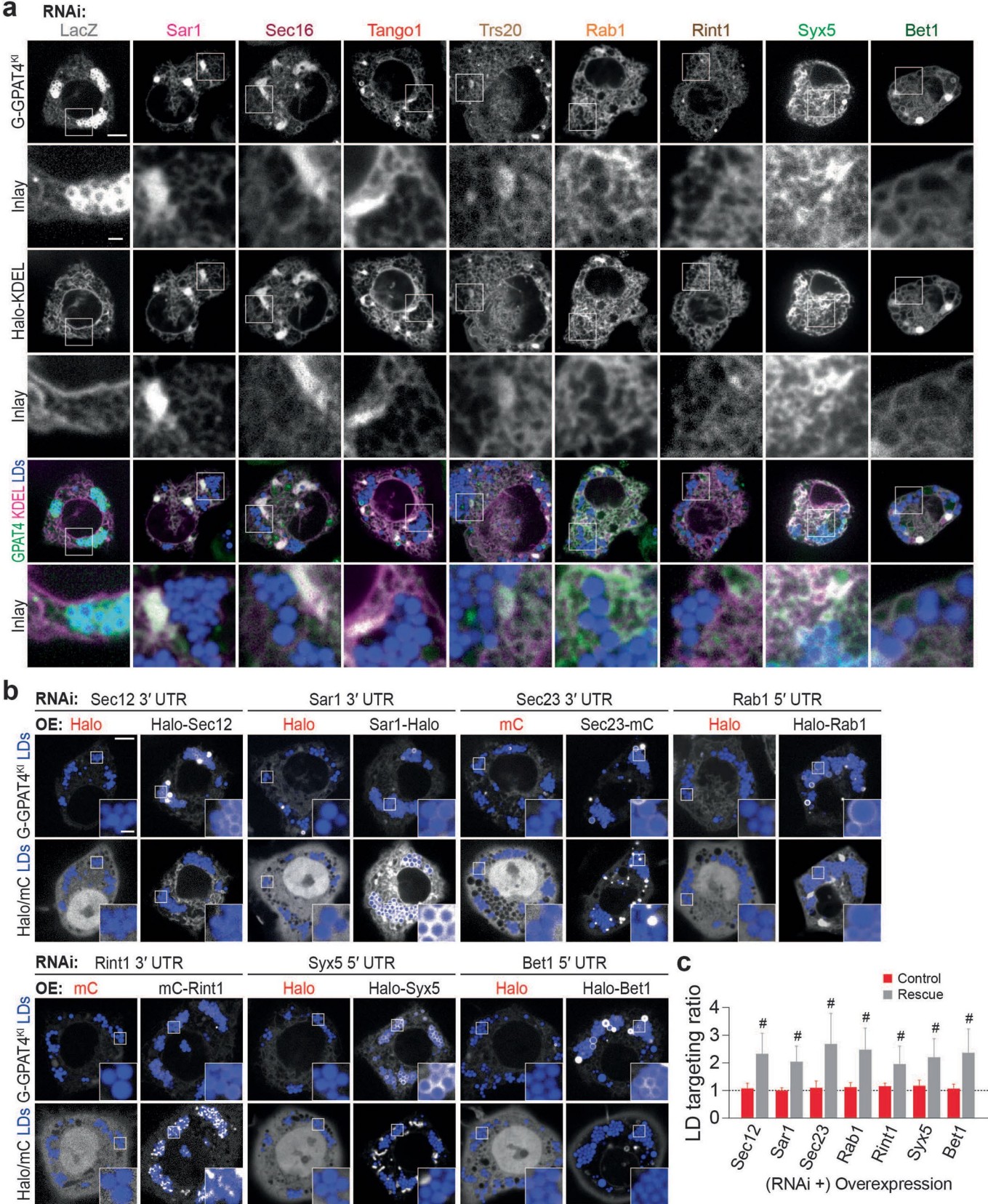

**Extended Data Fig. 3 | See next page for caption.**

**Extended Data Fig. 3 | GPAT4 remains in the ER when ERES components, membrane-fusion regulator, tether, and SNAREs are depleted. a**, Depletion of ERES components, Rab, tethering-complex components, and SNAREs results in endogenous GPAT4 co-localizing with ER marker Halo-KDEL without enrichment around LDs. Confocal imaging of live EGFP-GPAT4 endogenous knock-in cells upon RNAi of membrane fusion machinery components, transiently transfected with Halo-KDEL construct, 20 h after 1 mM oleic acid treatment. LDs were stained with MDH. Representative images from 3 independent experiments are shown. Scale bar, 5 and 1 μm (inlay). **b**, Effect of Sec12, Sar1, Sec23, Rab1, Rint1, Syx5, and Bet1 depletion on LD targeting of GPAT4 is rescued by expressing wildtype proteins. Confocal imaging of live EGFP-GPAT4 endogenous knock-in cells upon RNAi of Sec12, Sar1, Sec23, Rab1, Rint1, Syx5, and Bet1, followed by mCherry (mC) or Halo tagged constructs, 20 hr after 1 mM oleic acid treatment. Scale bar, 5 and 1 μm (inlay). **c**, Bar graph showing LD targeting ratios from the imaging experiment in **b**. Mean ± SD, n = (left to right) 30, 29, 27, 26, 26, 26, 42, 54, 29, 25, 31, 24, 32, 31 cells examined over 3 independent experiments. Two-tailed Student's t-test, #p < 0.0001, compared to respective control transfection. Source numerical data are available in source data.

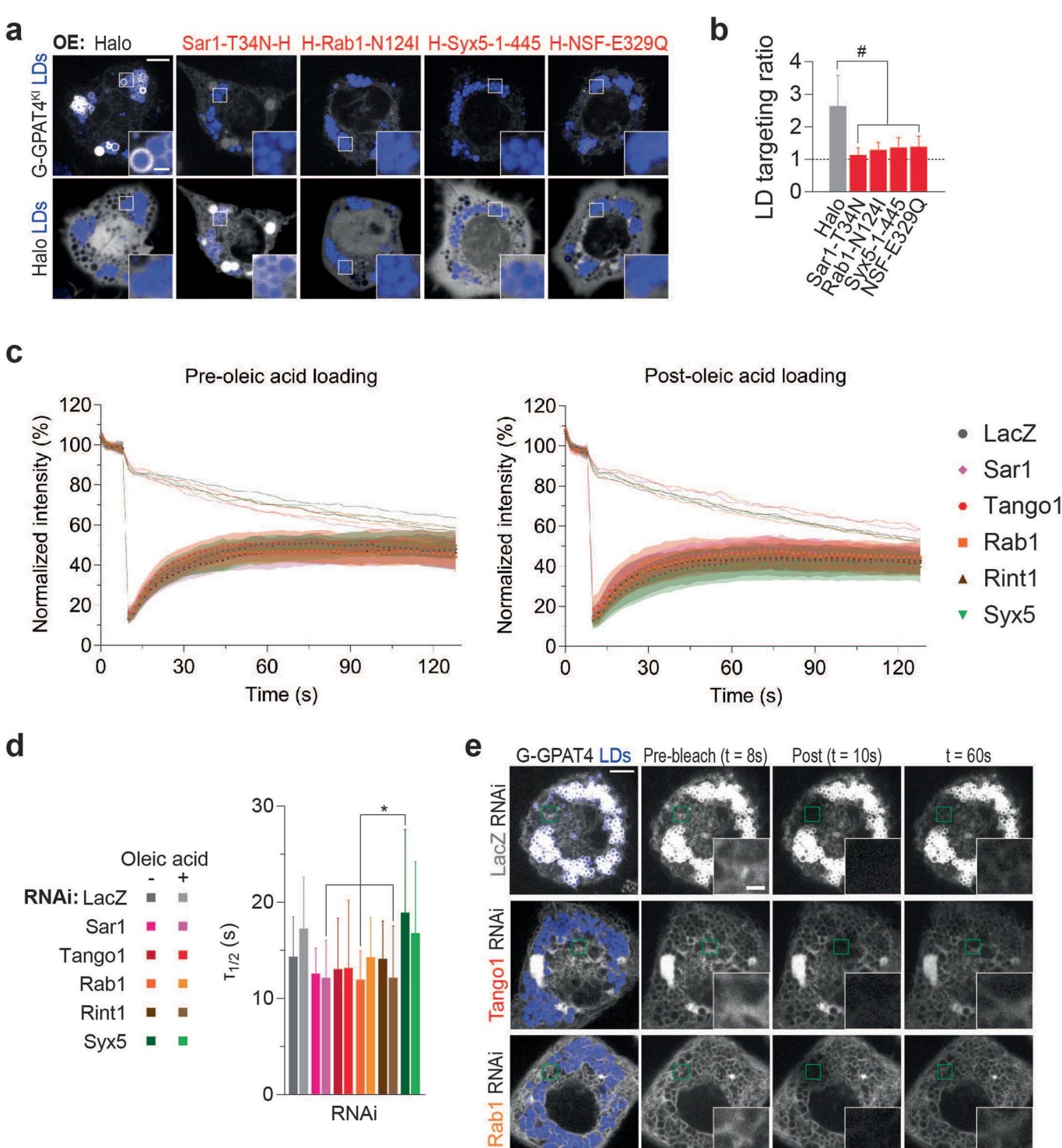

**Extended Data Fig. 4 | Depletion of ERES components, membrane-fusion regulator, tether, and SNAREs does not affect GPAT4 diffusion in the ER. a**, Expression of dominant-negative Sar1, Rab1, Syx5, and NSF mutants impairs GPAT4 targeting to LDs. Confocal imaging of live EGFP-GPAT4 endogenous knock-in cells upon transient transfection with Halo tagged constructs. H = Halo tag. Scale bar, 5 and 1 μm (inlay). **b**, Bar graph showing LD targeting ratios from the imaging experiment in **a**. Mean ± SD, n = (left to right) 60, 44, 46, 45, 40 cells examined over 3 independent experiments. One-way ANOVA with Bonferroni correction, #p < 0.0001. **c**, FRAP experiment of stably overexpressed EGFP-GPAT4 in the ER upon RNAi of LacZ, Sar1, Tango1, Rab1, Rint1, or Syx5 prior to or 20 h after 1 mM oleic acid treatment. Photobleaching at t = 8 s. Symbols indicate mean normalized intensities within the bleached region. SD is shown in transparent colors around the mean values. Solid lines above show mean normalized intensities outside the bleached region within the cell. **d**, Tau values for fluorescence recovery are comparable in all RNAi conditions pre- or post-oleic acid loading. Mean ± SD, n = 18 cells from 3 experiments. One-way ANOVA with Bonferroni correction, *p < 0.05 (from left to right: 0.0124, 0.0155, 0.0153). **e**, Representative images for the FRAP experiment in **c**, **d**. Green squares indicate the photobleached region. Scale bar, 5 and 1 μm (inlay). See also Supplementary Video 1. Source numerical data are available in source data.

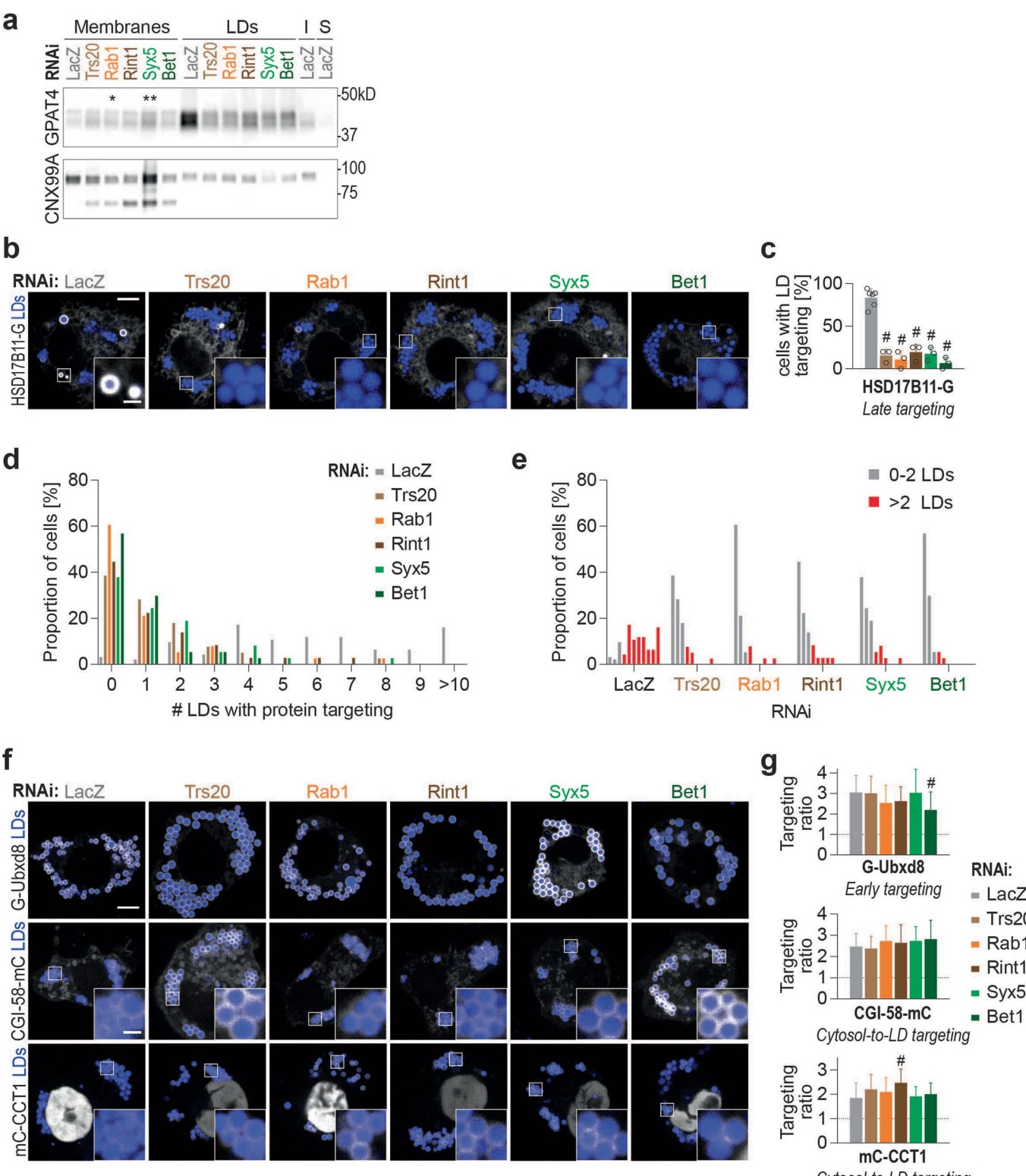

**Extended Data Fig. 5 | See next page for caption.**

**Extended Data Fig. 5 | Membrane-fusion regulator, tether, and SNAREs are required for late ER-to-LD targeting. a**, Western blot analysis of fractions of wildtype cells upon RNAi and LD induction. GPAT4 amounts remain the same or increase in membrane fractions but decrease in LD fractions upon depletion of membrane-fusion machinery components. Left: protein targets; right: ladder positions. I = Input (whole-cell lysate), S = soluble fraction. GPAT4 band intensities in membrane fractions: LacZ (1.00), Trs20 (2.25 ± 0.58), Rab1* (2.62 ± 1.04), Rint1 (2.11 ± 0.25), Syx5** (3.07 ± 0.40), and Bet1 (2.17 ± 0.28) (mean ± SD, n = 3). One-way ANOVA with Bonferroni correction, *p = 0.0455, **p = 0.0074, compared to LacZ. **b**, HSD17B11 requires membrane-fusion machinery components for LD targeting. Confocal imaging of live wildtype cells transiently transfected with HSD17B11-EGFP upon RNAi of fusion machinery, followed by a 20-hr incubation in oleate-containing medium. Scale bar, 5 and 1 μm (inlay). **c**, Bar graph showing percentage of cells with LD targeting (defined as those with >2 LDs with protein targeting) from the imaging experiment in **b**. Mean ± SD, n = 6 experiments for LacZ RNAi and 3 experiments for the rest (12–17 cells each). One-way ANOVA with Bonferroni correction, #p < 0.0001, compared to LacZ. **d, e**, Bar graph showing percentage of cells for a given number of LDs with HSD17B11-EGFP targeting upon RNAi of membrane-fusion machinery. Representative images in **b**. **f**, Depletion of specific Rab, membrane-tethering complex components, and SNAREs does not impair targeting of Ubxd8 (early ER-to-LD targeting) or CGI-58 and CCT1 (cytosol-to-LD targeting). Confocal imaging of live wildtype cells transiently transfected with EGFP- or mCherry-tagged constructs upon RNAi of fusion machinery, followed by a 20-hr incubation in oleate-containing medium. Scale bar, 5 and 1 μm (inlay). **g**, Bar graph showing targeting ratios from the imaging experiment in **f**. Mean ± SD, n = (left to right; top to bottom) 85, 35, 34, 36, 36, 34; 74, 33, 33, 31, 33, 31; 54, 36, 32, 33, 29, 32 cells examined over 3 independent experiments. For mCherry-CCT1, nuclear signal was excluded from the calculation. One-way ANOVA with Bonferroni correction, #p < 0.0001, compared to LacZ. Source numerical data and unprocessed blots are available in source data.

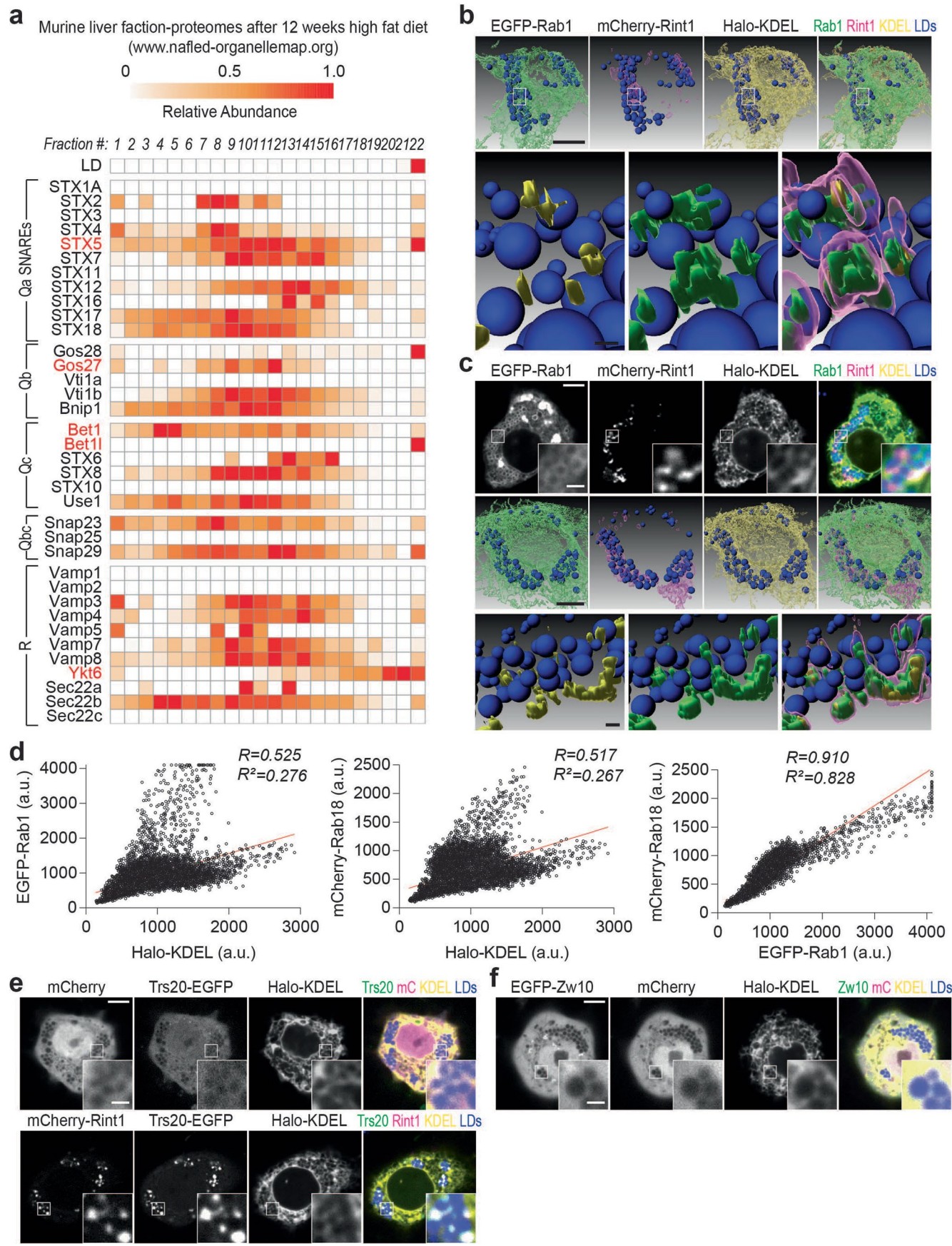

**Extended Data Fig. 6 | See next page for caption.**

**Extended Data Fig. 6 | Rab1, Rint1, and specific SNAREs associate with LDs. a**, Specific SNAREs are enriched in the LD fraction of murine fatty liver [data mined from nafld-organellemap.org[52]]. In this study, mice were subjected to 12 weeks of high-fat diet. Their livers were harvested and separated into 22 fractions using a sucrose gradient, and proteomes of the fractions were analyzed with mass spectrometry. First row shows the organellar migration pattern for LDs based on protein correlation profiling. SNAREs are classified according to their classes, and the predicted orthologs of SNAREs required for late ER-to-LD protein targeting (Syx5, membrin, Bet1, and Ykt6) are highlighted in red. **b**, Additional 3D reconstruction images for Fig. 3g, h. For the inlay images at the bottom, blue indicates LDs, yellow indicates regions of mCherry-Rint1 overlapping with ER marker (Halo-KDEL), green indicates regions of mCherry-Rint1 overlapping with LD surface and ER (EGFP-Rab1), and magenta indicates mCherry-Rint1. Scale bar, 5 and 1 μm (inlay). **c**, Additional example of 3D reconstruction experiment shown in **b**. Scale bar, 5 and 1 μm (inlay). **d**, Cytofluorograms between overexpressed tagged Rab1, Rab18, and KDEL (ER marker) for images shown in Fig. 3f. High R value suggests strong correlation between Rab1 and Rab18 intensities compared to between Rab1 and KDEL or between Rab18 and KDEL. $R^2$ values, as well as the regression line (red), are indicated. **e**, Overexpressed Rint1 recruits Trs20 to LDs. Localization of transiently co-transfected Trs20-EGFP and mCherry-Rint1 with respect to LDs. Representative images from 3 independent experiments are shown. Scale bar, 5 and 1 μm (inlay). **f**, Overexpressed Zw10 does not associate with LDs, in contrast to Rint1. Localization of transiently transfected EGFP-Zw10 with cytosolic (mCherry) and ER (Halo-KDEL) markers, with respect to LDs that were stained with MDH. Representative images from 3 independent experiments are shown. Scale bar, 5 and 1 μm (inlay). Source numerical data are available in source data.

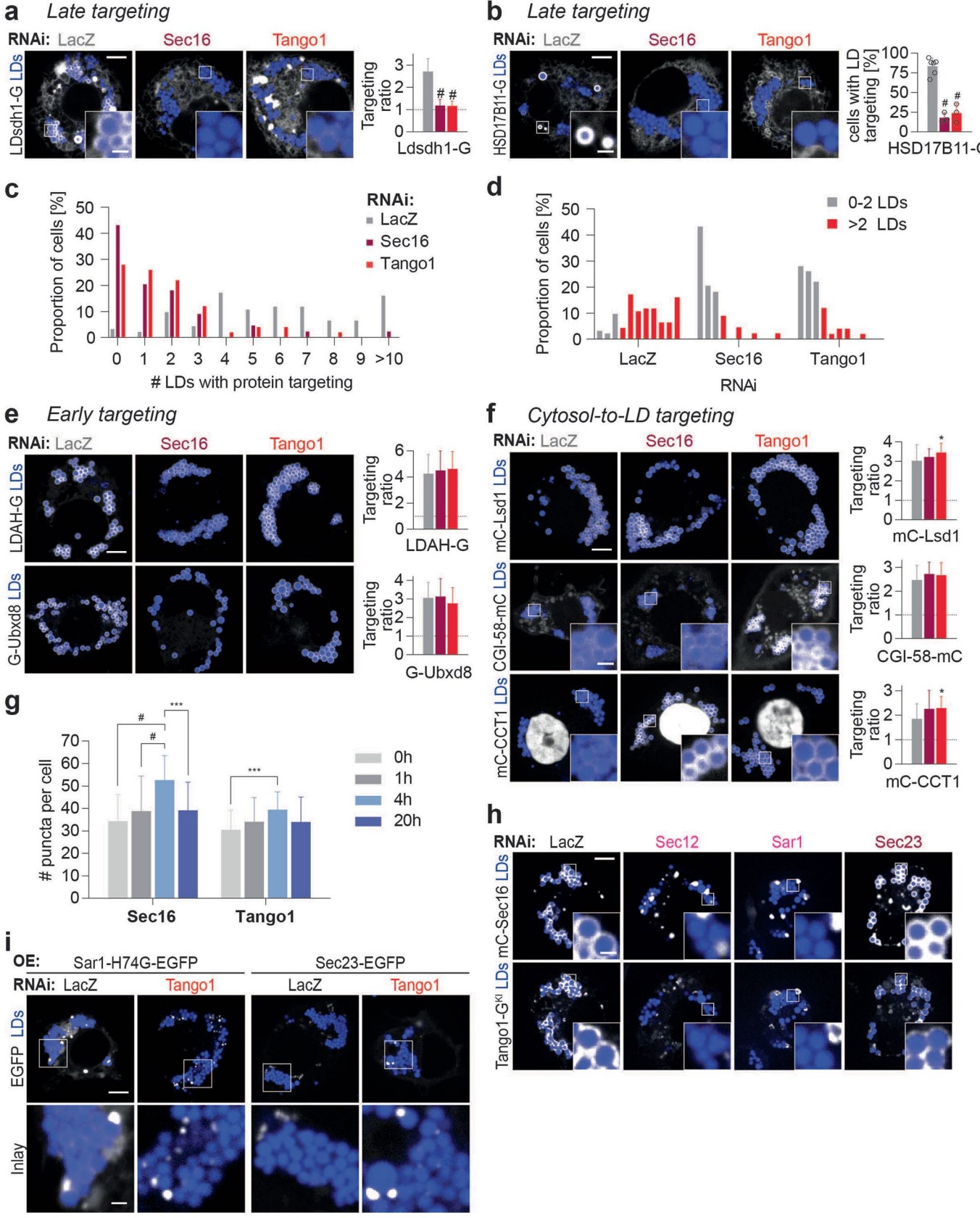

**Extended Data Fig. 7 | See next page for caption.**

**Extended Data Fig. 7 | ERES increase in number upon LD induction and are required for late ER-to-LD protein targeting. a**, Sec16 and Tango1 are required for LD targeting of Ldsdh1. Confocal imaging of wildtype cells upon RNAi of ERES components, followed by transient transfection with Ldsdh1-EGFP and a 20-hr incubation in oleate-containing medium. Scale bar, 5 and 1 μm (inlay). Mean ± SD, n = (left to right) 79, 41, 52 cells examined over 3 independent experiments. One-way ANOVA with Bonferroni correction, #p < 0.0001, compared to LacZ. **b**, Sec16 and Tango1 are required for LD targeting of HSD17B11. Scale bar, 5 and 1 μm (inlay). Bar graph shows percentage of cells with LD targeting (defined as those with >2 LDs with protein targeting). Mean ± SD, n = 3 experiments except 6 for LacZ RNAi (14–17 cells each). One-way ANOVA with Bonferroni correction, #p < 0.0001, compared to LacZ. **c, d**, Bar graph showing percentage of cells for a given number of LDs with HSD17B11-EGFP targeting upon RNAi of ERES components. Representative images in **b. e, f**, Sec16 and Tango1 are dispensable for LD targeting of (**e**) LDAH and Ubxd8 and (**f**) Lsd1, CGI-58, and CCT1. Scale bar, 5 and 1 μm (inlay). For mCherry-CCT1, nuclear signal was excluded from the calculation. Mean ± SD, n = (left to right) LDAH 87, 36, 36; Ubxd8 85, 44, 43; Lsd1 79, 39, 41; CGI-58 74, 36, 38; CCT1 54, 26, 31 cells examined over 3 independent experiments. One-way ANOVA with Bonferroni correction, *p < 0.05 (from top to bottom: 0.0159, 0.0193), compared to LacZ. **g**, LD induction transiently increases the number of Sec16 and Tango1 puncta. Representative images in Fig. 4e. Mean ± SD, n = (left to right) 38, 39, 33, 37; 38, 39, 35, 37 cells examined over 2 independent experiments. One-way ANOVA with Bonferroni correction, ***p < 0.001 (from left to right: 0.0001, 0.0008), #p < 0.0001. **h**, Localization of Sec16 around LDs and its recruitment of endogenous Tango1 require Sec12 and Sar1 but not Sec23. Representative images from 3 independent experiments are shown. Scale bar, 5 and 1 μm (inlay). **i**, Overexpressed Sar1-H74G and Sec23 accumulate around LDs upon Tango1 depletion. Representative images from 5 independent experiments are shown. Scale bar, 5 and 1 μm (inlay). Source numerical data are available in source data.

**a** Seipin KO cells: 0.5h in oleate containing medium

**b** RNAi:

**c** LD proteomics

**d**

**e**

Extended Data Fig. 8 | See next page for caption.

**Extended Data Fig. 8 | Late ER-to-LD protein targeting occurs independent of seipin, and LD proteomics reveal additional late targeting proteins.**
**a**, Seipin depletion allows for early targeting of late cargoes even in the absence of late ER-to-LD protein targeting machinery. Confocal imaging of live Seipin knock-out cells upon RNAi of ERES or fusion machinery components, followed by transient transfection with EGFP tagged constructs and a 0.5-hr incubation in oleate-containing medium. Scale bar, 5 and 1 μm (inlay). Quantification is shown in Fig. 6e. **b**, Depletion of late ER-to-LD targeting machinery does not affect association of seipin puncta with LDs. Confocal imaging of live GFP-seipin endogenous knock-in cells upon RNAi of ERES or fusion machinery components, followed by a 20-hr incubation in oleate-containing medium. Representative images from 3 independent experiments are shown. Scale bar, 5 and 1 μm (inlay). **c**, Heatmap of abundance of ERES and fusion-machinery components and seipin in LD fractions upon their depletion by RNAi (or gene deletion for Seipin) compared to LacZ control, as measured by mass spectrometry. **d**, ReepA targets LDs early, whereas LPCAT, ACSL5, and DHRS7B target LDs late upon LD induction. Confocal imaging of live wildtype cells transiently transfected with EGFP-tagged constructs at given timepoints after 1 mM oleic acid treatment. LDs were stained with MDH. Representative images are shown. Scale bar, 5 and 1 μm (inlay). **e**, Bar graph showing percentage of cells with LD targeting over time from the imaging experiment in **d**. Mean ± SD, n = 3 experiments (10–20 cells each). One-way ANOVA with Bonferroni correction, **p = 0.0043, ***p = 0.0009, #p < 0.0001. Source numerical data are available in source data.

# Reporting Summary

## Statistics

For all statistical analyses, confirm that the following items are present in the figure legend, table legend, main text, or Methods section.

| n/a | Confirmed | |
|---|---|---|
| ☐ | ☒ | The exact sample size ($n$) for each experimental group/condition, given as a discrete number and unit of measurement |
| ☐ | ☒ | A statement on whether measurements were taken from distinct samples or whether the same sample was measured repeatedly |
| ☐ | ☒ | The statistical test(s) used AND whether they are one- or two-sided<br>*Only common tests should be described solely by name; describe more complex techniques in the Methods section.* |
| ☒ | ☐ | A description of all covariates tested |
| ☐ | ☒ | A description of any assumptions or corrections, such as tests of normality and adjustment for multiple comparisons |
| ☐ | ☒ | A full description of the statistical parameters including central tendency (e.g. means) or other basic estimates (e.g. regression coefficient) AND variation (e.g. standard deviation) or associated estimates of uncertainty (e.g. confidence intervals) |
| ☐ | ☒ | For null hypothesis testing, the test statistic (e.g. $F$, $t$, $r$) with confidence intervals, effect sizes, degrees of freedom and $P$ value noted<br>*Give P values as exact values whenever suitable.* |
| ☒ | ☐ | For Bayesian analysis, information on the choice of priors and Markov chain Monte Carlo settings |
| ☒ | ☐ | For hierarchical and complex designs, identification of the appropriate level for tests and full reporting of outcomes |
| ☐ | ☒ | Estimates of effect sizes (e.g. Cohen's $d$, Pearson's $r$), indicating how they were calculated |

*Our web collection on statistics for biologists contains articles on many of the points above.*

## Software and code

Policy information about availability of computer code

| | |
|---|---|
| Data collection | NIS-Elements (4.51.01), Image Lab (6.0.1 build 34), IN Cell Analyzer 6000 Acquisition Software v1.0 |
| Data analysis | Fiji (ImageJ 2.3.0/1.53f51), GraphPad Prism 7, MATLAB 9.2, Imaris (9.8.2), MaxQuant (1.5.2.8), Perseus (1.5.1.6)<br>Original code (used for quantification of screen images) has been deposited at Harvard Dataverse and is publicly available (DOI: https://doi.org/10.7910/DVN/NKJDWS). |

For manuscripts utilizing custom algorithms or software that are central to the research but not yet described in published literature, software must be made available to editors and reviewers. We strongly encourage code deposition in a community repository (e.g. GitHub). See the Nature Portfolio guidelines for submitting code & software for further information.

## Data

Policy information about availability of data

All manuscripts must include a data availability statement. This statement should provide the following information, where applicable:
- Accession codes, unique identifiers, or web links for publicly available datasets
- A description of any restrictions on data availability
- For clinical datasets or third party data, please ensure that the statement adheres to our policy

Original screen images and quantification results are available at the Lipid Droplet Knowledge Portal (http://lipiddroplet.org/; currently available to reviewers through http://3.92.40.39/fly_gene). The mass spectrometry proteomics data have been deposited to the ProteomeXchange Consortium via the PRIDE partner repository with the dataset identifier PXD027283. UniProt Drosophila melanogaster database.

# Field-specific reporting

Please select the one below that is the best fit for your research. If you are not sure, read the appropriate sections before making your selection.

☒ Life sciences ☐ Behavioural & social sciences ☐ Ecological, evolutionary & environmental sciences

For a reference copy of the document with all sections, see nature.com/documents/nr-reporting-summary-flat.pdf

# Life sciences study design

All studies must disclose on these points even when the disclosure is negative.

| | |
|---|---|
| Sample size | No sample size calculation was performed. Genome-wide screen was repeated twice. Other experiments were repeated 2-3 independent times with 10-15 observations (cells) each. |
| Data exclusions | Outlier analysis was performed using ROUT method at Q = 1% on GraphPad Prism. |
| Replication | Number of independent experiments are indicated in the respective figure legends. No experiment was excluded in the analysis. |
| Randomization | The design of the RNAi assay plates for the genome-wide screen was randomized by the Drosophila RNAi Screening Center at Harvard Medical School. For all other cell experiments, randomization was not relevant/not performed, and the control and test conditions were performed side-by-side on the same day using the same reagents except for the treatment tested (such as RNAi or transfection). |
| Blinding | The RNAi assay plates for the genome-wide screen was prepared by the Drosophila RNAi Screening Center at Harvard Medical School, and the gene targets were not cross-referenced until all automatized analysis was completed. For all other cell experiments, blinding was not relevant/not performed. |

# Reporting for specific materials, systems and methods

We require information from authors about some types of materials, experimental systems and methods used in many studies. Here, indicate whether each material, system or method listed is relevant to your study. If you are not sure if a list item applies to your research, read the appropriate section before selecting a response.

## Materials & experimental systems

| n/a | Involved in the study |
|---|---|
| ☐ | ☒ Antibodies |
| ☐ | ☒ Eukaryotic cell lines |
| ☒ | ☐ Palaeontology and archaeology |
| ☒ | ☐ Animals and other organisms |
| ☒ | ☐ Human research participants |
| ☒ | ☐ Clinical data |
| ☒ | ☐ Dual use research of concern |

## Methods

| n/a | Involved in the study |
|---|---|
| ☒ | ☐ ChIP-seq |
| ☒ | ☐ Flow cytometry |
| ☒ | ☐ MRI-based neuroimaging |

## Antibodies

| | |
|---|---|
| Antibodies used | Rabbit anti-Drosophila GPAT4 (Dr. Tobias Walther, USA), Rat anti-Drosophila LDAH (Dr. Mathias Beller, Germany), Rabbit anti-Drosophila CCT1 (Dr. Tobias Walther, USA), Mouse anti-Drosophila CNX99A (Developmental Studies Hybridoma Bank, Cat# Cnx99A 6-2-1), Mouse anti-α-tubulin (Sigma Aldrich, Cat# T5168), Rabbit anti-Drosophila Sec16 (Dr. Catherine Rabouille, Netherlands), Guinea pig anti-Drosophila Tango1 (Dr. Sally Horne-Badovinac, USA), Mouse anti-rabbit IgG-HRP (Santa Cruz Biotechnology, Cat# sc-2357), Mouse anti-IgG kappa binding protein-HRP (Santa Cruz Biotechnology, Cat# sc-516102), Goat anti-rat IgG H&L-HRP (Abcam, Cat# ab97057), Alexa Fluor 647 goat anti-rabbit IgG (Thermo Scientific, Cat# A-21244), Alexa Fluor 488 goat anti-guinea pig IgG (Thermo Scientific, Cat# A-11073) |
| Validation | Rabbit anti-Drosophila GPAT4 (Wilfling et al., Dev Cell, 2013; validated with RNAi in Drosophila cells), Rat anti-Drosophila LDAH (Thiel et al., JCS, 2013; validated with RNAi in Drosophila cells), Rabbit anti-Drosophila CCT1 (Krahmer et al., Cell Metab, 2011; validated with RNAi in Drosophila cells), Mouse anti-Drosophila CNX99A (Riedel et al., Biology Open, 2016; validated with RNAi in Drosophila cells), Mouse anti-α-tubulin (Sigma Aldrich, Cat# T5168; Independent Antibody Verification – Demonstrating antibody specificity through the use of multiple antibodies against target in IHC or ICC), Rabbit anti-Drosophila Sec16 (Ivan et al., MBoC, 2008; validated with RNAi in Drosophila cells), Guinea pig anti-Drosophila Tango1 (Lerner et al., Dev Cell, 2013; validated with RNAi in Drosophila), Mouse anti-rabbit IgG-HRP (Santa Cruz Biotechnology, Cat# sc-2357; validated with western blot), Mouse anti-IgG kappa binding protein-HRP (Santa Cruz Biotechnology, Cat# sc-516102; validated with western blot), Goat anti-rat IgG H&L-HRP (Abcam, Cat# ab97057; validated with western blot), Alexa Fluor 647 goat anti-rabbit IgG (Thermo Scientific, Cat# A-21244; validated with immunofluorescence), Alexa Fluor 488 goat anti-guinea pig IgG (Thermo Scientific, Cat# A-11073; validated with immunofluorescence) |

# Eukaryotic cell lines

Policy information about cell lines

| | |
|---|---|
| Cell line source(s) | Drosophila S2R+ cells used in the study were obtained from Dr. Norbert Perrimon (Harvard Medical School) and can be purchased from Drosophila Genomics Resource Center (stock number 150). Additional cell lines were created using CRISPR-Cas9 system as noted in the methods (endogenous EGFP knock-in cells to GPAT4 locus; seipin knockout cells) |
| Authentication | S2R+ cells were not authenticated. The generated cell lines were verified with western blot and mass spectrometer-based proteomics. |
| Mycoplasma contamination | All cell lines tested negative for mycoplasma. |
| Commonly misidentified lines (See ICLAC register) | No commonly misidentified cell lines were used. |

