## [Peer Review File · Nature Cell Biology]

Peer Review Information

Journal: Nature Cell Biology

Manuscript Title: Identification of two pathways mediating protein targeting from ER to lipid droplets

Corresponding author names: Robert V. Farese, Jr., Tobias C. Walther

Reviewer Comments & Decisions:

Decision Letter, initial version:

Subject: Decision on Nature Cell Biology submission NCB-S46664

Message: *Please delete the link to your author homepage if you wish to forward this email to co-authors.

Dear Tobi,

Thank you for submitting your manuscript, "Identification of two pathways mediating protein targeting from ER to lipid droplets", to Nature Cell Biology and thank you for your patience with the peer review process. It has now been seen by 3 referees, who are experts in LDs (referee 1); membrane protein biosynthesis at the ER/ER/protein trafficking (referee 2); and LDs/lipid dynamics (referee 3). As you will see from their comments (attached below), they find this work of potential interest, but have raised substantial concerns, which in our view would need to be addressed with considerable revisions before we can consider publication in Nature Cell Biology.

As per standard practice at the journal, we have discussed the comments from the reviewers in detail within the editorial team, including with our chief editor, to identify key referee points that should be addressed with priority to strengthen the core new results, as opposed to requests that are beyond the scope of the current study. To guide the scope of the revisions, I have listed these points below. We are committed to providing a fair and constructive peer-review process, so please feel free to contact me if you would like to discuss any of the referee comments further. Our typical revision period is six months, however, please do not hesitate to contact me if you anticipate any delays or issues addressing the reviews. We are happy to discuss further as needed.

In our view, for reconsideration at NCB, it would be essential to dedicate efforts in revision as follows:

1. Two reviewers commented on the limited understanding of the ways in which the screen hits function in ER-to-LD protein targeting and that whether they function in the same pathway was not definitively proven. Expanding the mechanistic understanding of the hits would greatly enhance the study, and to elaborate on the current dataset, Rev#1 suggested assessing whether the hits do compromise membrane bridge formation/maintenance, as previously visualized in other work from your lab, without visualizing the dynamics of the bridge or brief/transient fusion events. As the bridges are an important part of your overall model (they're speculatively mentioned in the abstract as well), we agree that attempts to visualize them in the depletion conditions could provide some more clarity on the impact of the ERTOLD mediators on protein targeting during and after LD biogenesis and could enhance the mechanistic understanding of the pathways, and therefore would go a long way in addressing Rev#1 and Rev#3's comments.

2. The reviewers requested controls (e.g., for depletion studies) that are important as well as more clarity on the localization of GPAT4 (Rev#1, Rev#2) and Rab1/Rin1 (Rev#2 and Rev#3) – we agree with the reviewers that more clarity would be beneficial.

3. All other referee concerns pertaining to methodological details, clarifications of existing data, and textual changes should be addressed as appropriate.

4. Finally, please pay close attention to our guidelines on statistical and methodological reporting (listed below) as failure to do so may delay the reconsideration of the revised manuscript. In particular please provide:

We would be happy to consider a revised manuscript that would satisfactorily address these points, unless a similar paper is published elsewhere, or is accepted for publication in Nature Cell Biology in the meantime.

2When revising the manuscript please:

- ensure that it conforms to our format instructions and publication policies (see below and www.nature.com/nature/authors/).
- provide a point-by-point rebuttal to the full referee reports verbatim, as provided at the end of this letter.
- provide the completed Editorial Policy Checklist (found here <https://www.nature.com/authors/policies/Policy.pdf>), and Reporting Summary (found here <https://www.nature.com/authors/policies/ReportingSummary.pdf>). This is essential for reconsideration of the manuscript and these documents will be available to editors and referees in the event of peer review. For more information see <http://www.nature.com/authors/policies/availability.html> or contact me.

[REDACTED]

We hope that you will find our referees' comments and editorial guidance helpful. Please do not hesitate to contact me if there is anything you would like to discuss. Thank you very much for considering NCB for your work.

Best wishes,

Melina

Melina Casadio, PhD
Senior Editor, Nature Cell Biology
ORCID ID: <https://orcid.org/0000-0003-2389-2243>

Reviewers' Comments:

Reviewer #1:

Remarks to the Author:

Lipid droplets are ubiquitous neutral lipid storage organelles that are derived from the ER through a process involving the deposition of neutral lipids and budding of the droplet from the outer leaflet of the ER bilayer. Proteins decorating the surface of the lipid droplet are from two general classes: 1) Class I (or type I) lipid droplet proteins insert first into the ER and then traffic to lipid droplets. These proteins tend to have hydrophobic hairpins, but how they traffic to and concentrate in lipid droplets is still poorly understood. 2) Class II (or type II) lipid droplet proteins insert directly into lipid droplets, often employing amphipathic helices that recognize distinct features of lipid droplet membranes.

Here, the authors examine the mechanism for class I lipid droplet protein targeting. Their findings indicate that there are two types of class I lipid droplet proteins that differ in their trafficking kinetics and pathway to lipid droplets. Their primary focus is on the late class I lipid droplet proteins that traffic to already formed lipid droplets. Employing a fluorescence-based reporter of late class I lipid droplet protein trafficking, they perform a genome-wide siRNA screen to identify machinery involved in this pathway. This approach identifies a set of regulators, including membrane-fusion machinery, tethers, and SNARE proteins, that establishes a putative trafficking pathway for late class I lipid droplet proteins. This pathway may involve membrane bridges, but this remains speculative. They also devise a clever proteomics approach to define early and late class I lipid droplet proteins. Interestingly, loss of seipin results in early targeting of late class I lipid droplet proteins.

Overall, this is a well-executed study that seeks to answer a challenging question in the field – how do proteins traffic to lipid droplets? Their data represent an important advance in the field and support an intriguing model in which there are two distinct trafficking pathways for class I lipid droplet proteins: an early pathway in which proteins traffic to forming lipid droplets and a late pathway in which proteins traffic to already formed lipid droplets, likely due to exclusion by seipin at the ER-lipid droplet membrane contact and possibly involving membrane bridges.

Major comments

1) I am strongly against introducing the terms ERTOLD and CYTOLD. This seems unnecessary. The authors already introduce a framework for classifying lipid droplet proteins based on their localization origin and trafficking pathway (<https://doi.org/10.1016/j.tcb.2016.02.007>). In Kory et al., they introduce class I and II lipid droplet proteins, stating “Class I proteins often have a membrane-embedded, hydrophobic 'hairpin' motif, and access LDs from the endoplasmic reticulum (ER) either during LD formation or after formation via ER-LD membrane bridges. Class II proteins access the LD surface from the cytosol and bind through amphipathic helices or other hydrophobic domains.” This nomenclature is sufficient without introducing ERTOLD and CYTOLD. Similarly, other fields don't use ERTOG (ER to Golgi), ERTOP (ER to peroxisome), etc.

2) While the manuscript nicely validates candidate genes identified in the screen, the conclusion that they function within the same pathway and that they actually act by regulating membrane bridges is preliminary. Additional mechanistic data supporting this model would greatly enhance the manuscript. These bridges were previously visualized by fluorescence and EM imaging (<https://doi.org/10.1016/j.devcel.2013.01.013>).

Additional comments

1) A large number of genes were identified as regulators of GPAT4 targeting (1,100 genes). The rationale for selecting the handful of membrane / trafficking related proteins is clear, but it seems like additional discussion of the other 1,000 genes is warranted. Do 1,100 genes really influence GPAT4 trafficking? What portion are false positives? Are genes that influence lipid droplet size and number controlled for?

2) Do the newly identified late class I lipid droplet proteins show similar temporal kinetics of lipid droplet trafficking?

3) Where is GPAT4 localizing when it can't traffic to the LD? GPAT4 appears to be co-localizing at the ER in Figure 1a, but it is unclear in some panels showing microscopy of GPAT4 in the knockdowns. For example, there is barely any GPAT4 signal in Fig 3a. Is it still ER localized or does it fail to insert into the ER?

4) It is unclear if these are true knockdowns (KDs) since there is no validation by Western or any other method. It would be nice to see KD validation for their top hits.

Reviewer #2:

Remarks to the Author:

This study has investigated the pathways by which proteins populate lipid droplets (LDs). Earlier work has established that there are multiple routes, including direct cytosol-to-LD (CYTOLD) and ER-to-LD (ERTOLD) pathways. It was also suggested based on kinetic analysis that the ERTOLD substrates arrive at LDs at different times. The current work provides considerable clarity to the pathways of ERTOLD. The authors document nicely that there is an early and a late ERTOLD pathway that differ in their requirements. Using a screening strategy, they go on to find a set of genes required for late ERTOLD, place these into a sensible working framework, and show how this pathway is different from the early ERTOLD pathway. Strikingly, it emerges that the early ERTOLD pathway restricts certain proteins from going to LDs, and these are later targeted via the late pathway. Removing the restriction factor (seipin) now allows the late clients to enter LDs early, and importantly, renders the genes of the late pathway dispensable for LD targeting.

The work is terrific in every way: clear and convincing data, well-written, biologically important, and sensibly interpreted and discussed. It opens up numerous mechanistic questions about the nature of ER exit sites and how they might link up to LDs, and what each of the late ERTOLD pathway factors contribute to the process. I strongly support its publication and have only a few minor suggestions for improvement. Note that any experimental suggestions in my comments below should be considered optional, and I ask only out of curiosity.

Minor comments:

1) Perhaps the most intriguing question that might be worth a couple of lines in the Discussion is why the cell would want to temporally segregate LD clients into early and late. In other words, is there a logic to why early clients would be needed first (or conversely, why late clients should be restricted until later)? Alternatively, is there some shared feature of late clients that make them unsuitable for the early pathway? I feel readers of this study might want to know what the authors are thinking.

2) It seems to me that forming a bridge between the ERES and LD for the late pathway might depend on an LD protein that is already there. An attractive idea is if an early ERTOLD client actually participates in the late ERTOLD pathway. Is there any evidence for this, perhaps in earlier observations? For example, does eliminating any early clients impact late ERTOLD clients?

3) It is stated that endogenous GPAT4 is decreased in the LD fraction and increased in the microsomal fraction “for each of these factor knockdowns” (referring to Trs20, Rab1, Rint1, Syx5 or Bet1). The

6decrease in LD is clear. The increase in the membrane fraction in Fig. S3a is less clear and is statistically significant for only Rab1 and Syx5. Perhaps re-phrase or clarify.

4) In some of the knockdowns that impact LD localisation, the non-localised protein shows some bright puncta (e.g., Rint1 in Fig. 3D). Perhaps a comment about this is worthwhile?

5) It is a little hard to tell in Fig. 3F whether the overexpressed Rab1 is really on the LDs or just occupying the space between them. If available, an image with a different overexpressed Rab protein might be helpful for comparison, and would also help convince the reader about the specificity. This is not essential as there is separate evidence for Rab1 being involved; so an optional suggestion for the authors to consider.

Reviewer #3:

Remarks to the Author:

The manuscript of Song et al addresses the targeting pathways of membrane proteins from the ER to lipid droplets (LDs). LDs are formed and mature from the ER membrane and require several metabolic and structural proteins for their function as dynamic fatty acid and lipid depots. How these proteins traffic to and from the LD surface remains incompletely understood. The authors performed a genome-wide screen in fly cells expressing a reporter ER membrane protein that is known to accumulate on LDs with slow kinetics. The screen identified an enrichment of membrane trafficking components. In follow-up analyses, the authors validated the requirement of these components such as Rab1, Snares, a TRAPP subunit and ER exit site machinery and further showed that removal of seipin rescues these targeting defects, consistent with a role of seipin in controlling traffic at ER-LD contact sites. Finally, membrane proteins that use this ER to LD targeting were identified.

Overall, the screen and follow-up experiments are well executed (but see below for some limitations on imaging/validation) and I find the topic very interesting. The paper is well written. The authors demonstrate a clear requirement for certain trafficking factors in the protein targeting from the ER to LDs. However, my view is that the study does not significantly extend beyond reporting and characterizing the screen hits. Previous studies, including from the authors, have established a link between trafficking factors and LDs but the underlying basis remains unclear. It would be important for the authors to at least test their main hypothesis of a membrane fusion reaction at the LD site required for ER to LD traffic. It would be also important to confirm that inhibition of ER to Golgi traffic, which can affect ER membrane, does not have an indirect impact on GPAT4 movement, especially under conditions

7of fatty acid overload. Finally it is not clear to me why membrane bridges connecting the GPAT4-containing LDs and the ER would necessarily need a membrane fusion event as opposed to structural rearrangement and/or tubulation of the ER membrane.

Other comments:

An argument for the specificity of the pathway is the localization of membrane fusion factors and ER exit site machinery to LDs. I think these results could be strengthened. For example for the localization of Rab1 and Rint1 to LDs (Fig3f), one wonders whether the distribution of Rint1 corresponds to an ER subdomain in the vicinity of LDs rather than LDs. This also applies to the Tango/Sec16 localization (Fig. 4e, f,g). Higher resolution methods would be beneficial here. Alternatively, how would these localizations compare to other ER resident and trafficking components following the addition of oleate?

Most of the membrane trafficking hits from the screen that have been followed up have not been validated by confirming their protein levels - could be this a reason as to why components of the same trafficking complexes show different phenotypes? Also, for positive hits with the exception of Rab1, there are no rescue experiments.

I am not sure I understand the statement of a machinery that "...supports *heterotypic* organelle fusion of the ER and LDs" in the introduction and discussion. LDs are derived from the ER and essentially constitute a specialized ER domain rather than an organelle with a separate identity.

READABILITY OF MANUSCRIPTS – Nature Cell Biology is read by cell biologists from diverse backgrounds, many of whom are not native English speakers. Authors should aim to communicate their findings clearly, explaining technical jargon that might be unfamiliar to non-specialists, and avoiding non-standard abbreviations. Titles and abstracts should concisely communicate the main findings of the

8study, and the background, rationale, results and conclusions should be clearly explained in the manuscript in a manner accessible to a broad cell biology audience. Nature Cell Biology uses British spelling.

FINANCIAL AND NON-FINANCIAL COMPETING INTERESTS – the authors must include one of three declarations: (1) that they have no financial and non-financial competing interests; (2) that they have financial and non-financial competing interests; or (3) that they decline to respond, after the Author Contributions section. This statement will be published with the article, and in cases where financial and non-financial competing interests are declared, these will be itemized in a web supplement to the

article. For further details please see <https://www.nature.com/licenceforms/nrg/competing-interests.pdf>.

Methods should be written concisely, but should contain all elements necessary to allow interpretation and replication of the results. As a guideline, Methods sections typically do not exceed 3,000 words. The Methods should be divided into subsections listing reagents and techniques. When citing previous methods, accurate references should be provided and any alterations should be noted. Information must be provided about: antibody dilutions, company names, catalogue numbers and clone numbers for monoclonal antibodies; sequences of RNAi and cDNA probes/primers or company names and catalogue numbers if reagents are commercial; cell line names, sources and information on cell line identity and authentication. Animal studies and experiments involving human subjects must be reported in detail, identifying the committees approving the protocols. For studies involving human subjects/samples, a statement must be included confirming that informed consent was obtained. Statistical analyses and information on the reproducibility of experimental results should be provided in a section titled “Statistics and Reproducibility”.

All Nature Cell Biology manuscripts submitted on or after March 21 2016 must include a Data availability statement at the end of the Methods section. For Springer Nature policies on data availability see <http://www.nature.com/authors/policies/availability.html>; for more information on this particular policy see <http://www.nature.com/authors/policies/data/data-availability-statements-data-citations.pdf>. The Data availability statement should include:

- Accession codes for primary datasets (generated during the study under consideration and designated as "primary accessions") and secondary datasets (published datasets reanalysed during the study under consideration, designated as "referenced accessions"). For primary accessions data should be made public to coincide with publication of the manuscript. A list of data types for which submission to community-endorsed public repositories is mandated (including sequence, structure, microarray, deep sequencing data) can be found here <http://www.nature.com/authors/policies/availability.html#data>.
- Unique identifiers (accession codes, DOIs or other unique persistent identifier) and hyperlinks for datasets deposited in an approved repository, but for which data deposition is not mandated (see here for details <http://www.nature.com/sdata/data-policies/repositories>).
- At a minimum, please include a statement confirming that all relevant data are available from the authors, and/or are included with the manuscript (e.g. as source data or supplementary information), listing which data are included (e.g. by figure panels and data types) and mentioning any restrictions on availability.
- If a dataset has a Digital Object Identifier (DOI) as its unique identifier, we strongly encourage including this in the Reference list and citing the dataset in the Methods.

We recommend that you upload the step-by-step protocols used in this manuscript to the Protocol Exchange. More details can found at www.nature.com/protocolexchange/about.

All imaging data should be accompanied by scale bars, which should be defined in the legend. Cropped images of gels/blots are acceptable, but need to be accompanied by size markers, and to retain visible background signal within the linear range (i.e. should not be saturated). The boundaries of panels with low background have to be demarked with black lines. Splicing of panels should only be considered

if unavoidable, and must be clearly marked on the figure, and noted in the legend with a statement on whether the samples were obtained and processed simultaneously. Quantitative comparisons between samples on different gels/blots are discouraged; if this is unavoidable, it should only be performed for samples derived from the same experiment with gels/blots were processed in parallel, which needs to be stated in the legend.

- For line art, graphs, charts and schematics we prefer Adobe Illustrator (.AI), Encapsulated PostScript (.EPS) or Portable Document Format (.PDF). Files should be saved or exported as such directly from the application in which they were made, to allow us to restyle them according to our journal house style.
- We accept PowerPoint (.PPT) files if they are fully editable. However, please refrain from adding PowerPoint graphical effects to objects, as this results in them outputting poor quality raster art. Text used for PowerPoint figures should be Helvetica (preferred) or Arial.
- We do not recommend using Adobe Photoshop for designing figures, but we can accept Photoshop generated (.PSD or .TIFF) files only if each element included in the figure (text, labels, pictures, graphs, arrows and scale bars) are on separate layers. All text should be editable in 'type layers' and line-art such as graphs and other simple schematics should be preserved and embedded within 'vector smart objects' - not flattened raster/bitmap graphics.
- Some programs can generate Postscript by 'printing to file' (found in the Print dialogue). If using an application not listed above, save the file in PostScript format or email our Art Editor, Allen Beattie for advice (a.beattie@nature.com).

Unprocessed scans of all key data generated through electrophoretic separation techniques need to be presented in a supplementary figure that should be labelled and numbered as the final supplementary

figure, and should be mentioned in every relevant figure legend. This figure does not count towards the total number of figures and is the only figure that can be displayed over multiple pages, but should be provided as a single file, in PDF or TIFF format. Data in this figure can be displayed in a relatively informal style, but size markers and the figures panels corresponding to the presented data must be indicated.

The total number of Supplementary Figures (not including the “unprocessed scans” Supplementary Figure) should not exceed the number of main display items (figures and/or tables (see our Guide to Authors and March 2012 editorial <http://www.nature.com/ncb/authors/submit/index.html#suppinfo>; <http://www.nature.com/ncb/journal/v14/n3/index.html#ed>). No restrictions apply to Supplementary Tables or Videos, but we advise authors to be selective in including supplemental data.

GUIDELINES FOR EXPERIMENTAL AND STATISTICAL REPORTING

REPORTING REQUIREMENTS – To improve the quality of methods and statistics reporting in our papers we have recently revised the reporting checklist we introduced in 2013. We are now asking all life sciences authors to complete two items: an Editorial Policy Checklist (found here <https://www.nature.com/authors/policies/Policy.pdf>) that verifies compliance with all required editorial policies and a reporting summary (found here <https://www.nature.com/authors/policies/ReportingSummary.pdf>) that collects information on experimental design and reagents. These documents are available to referees to aid the evaluation of the manuscript. Please note that these forms are dynamic ‘smart pdfs’ and must therefore be downloaded and completed in Adobe Reader. We will then flatten them for ease of use by the reviewers. If you would like to reference the guidance text as you complete the template, please access these flattened versions at <http://www.nature.com/authors/policies/availability.html>.

STATISTICS – Wherever statistics have been derived the legend needs to provide the n number (i.e. the sample size used to derive statistics) as a precise value (not a range), and define what this value represents. Error bars need to be defined in the legends (e.g. SD, SEM) together with a measure of centre (e.g. mean, median). Box plots need to be defined in terms of minima, maxima, centre, and

percentiles. Ranges are more appropriate than standard errors for small data sets. Wherever statistical significance has been derived, precise p values need to be provided and the statistical test used needs to be stated in the legend. Statistics such as error bars must not be derived from $n < 3$. For sample sizes of $n < 5$ please plot the individual data points rather than providing bar graphs. Deriving statistics from technical replicate samples, rather than biological replicates is strongly discouraged. Wherever statistical significance has been derived, precise p values need to be provided and the statistical test stated in the legend.

Author Rebuttal to Initial comments

Identification of two pathways mediating protein targeting from ER to lipid droplets (NCB-S46664)

Point-by point response to the reviewers:

Reviewer #1:

Remarks to the Author:

15Lipid droplets are ubiquitous neutral lipid storage organelles that are derived from the ER through a process involving the deposition of neutral lipids and budding of the droplet from the outer leaflet of the ER bilayer. Proteins decorating the surface of the lipid droplet are from two general classes: 1) Class I (or type I) lipid droplet proteins insert first into the ER and then traffic to lipid droplets. These proteins tend to have hydrophobic hairpins, but how they traffic to and concentrate in lipid droplets is still poorly understood. 2) Class II (or type II) lipid droplet proteins insert directly into lipid droplets, often employing amphipathic helices that recognize distinct features of lipid droplet membranes.

Here, the authors examine the mechanism for class I lipid droplet protein targeting. Their findings indicate that there are two types of class I lipid droplet proteins that differ in their trafficking kinetics and pathway to lipid droplets. Their primary focus is on the late class I lipid droplet proteins that traffic to already formed lipid droplets. Employing a fluorescence-based reporter of late class I lipid droplet protein trafficking, they perform a genome-wide siRNA screen to identify machinery involved in this pathway. This approach identifies a set of regulators, including membrane-fusion machinery, tethers, and SNARE proteins, that establishes a putative trafficking pathway for late class I lipid droplet proteins. This pathway may involve membrane bridges, but this remains speculative. They also devise a clever proteomics approach to define early and late class I lipid droplet proteins. Interestingly, loss of seipin results in early targeting of late class I lipid droplet proteins.

Overall, this is a well-executed study that seeks to answer a challenging question in the field – how do proteins traffic to lipid droplets? Their data represent an important advance in the field and support an intriguing model in which there are two distinct trafficking pathways for class I lipid droplet proteins: an early pathway in which proteins traffic to forming lipid droplets and a late pathway in which proteins traffic to already formed lipid droplets, likely due to exclusion by seipin at the ER-lipid droplet membrane contact and possibly involving membrane bridges.

We thank the reviewer for her/his constructive comments and for raising important points. We performed extensive new experiments that led to exciting insights and greatly improved our manuscript.

Major comments

1) I am strongly against introducing the terms ERTOLD and CYTOLD. This seems unnecessary. The authors already introduce a framework for classifying lipid droplet proteins based on their localization origin and trafficking pathway (<https://doi.org/10.1016/j.tcb.2016.02.007>). In Kory et al., they introduce class I and II lipid droplet proteins, stating “Class I proteins often have a membrane-embedded, hydrophobic 'hairpin' motif, and access LDs from the endoplasmic reticulum (ER) either during LD formation or after formation via ER-LD membrane bridges. Class II proteins access the LD surface from the cytosol and bind through amphipathic helices or other hydrophobic domains.” This nomenclature is sufficient without introducing ERTOLD and

16CYTOLD. Similarly, other fields don't use ERTOG (ER to Golgi), ERTOP (ER to peroxisome), etc.

We thank the reviewer for this opinion. As the reviewer points out, our laboratory previously classified proteins into two classes of proteins that target LDs¹. An issue we encountered is that class I versus class II is not intuitive, and investigators, including ourselves, often mix up the two classes. We therefore introduced acronyms that describe the process for “ER-to-LD targeting” (ERTOLD) and “cytoplasm-to-LD targeting” (CYTOLD) in a recent review². Nevertheless, we revised the manuscript to exclude this new terminology in the current paper and will let the field decide whether to adopt this nomenclature in the future.

2) While the manuscript nicely validates candidate genes identified in the screen, the conclusion that they function within the same pathway and that they actually act by regulating membrane bridges is preliminary. Additional mechanistic data supporting this model would greatly enhance the manuscript. These bridges were previously visualized by fluorescence and EM imaging (<https://doi.org/10.1016/j.devcel.2013.01.013>).

We thank the reviewer for this comment and agree. We added substantial new data that strengthen our conclusions.

As the reviewer points out, our laboratory previously reported ER-LD bridges by electron tomography. It therefore is logical to ask whether these bridges depend on the factors we identify. Although we performed extensive EM experiments to address this, we encountered three confounding factors that make this experiment next to impossible. i) Although abundant on a subset of LDs³, bridges are still rare in electron micrographs of EM sections (because they only sample a very small section of a small percentage of the LDs), thereby complicating the quantitation of their absence. ii) Even in the absence of bridge factors, seipin-mediated ER-LD connections persist (Supplementary Fig. 8b**), thus complicating analysis by EM. iii) Electron tomography, which is better suited to visualize bridges, does not allow the collection of enough data to make a statistically sound argument. We edited the text to better explain these issues.**

An alternative approach is to visualize the ER-LD bridges with live-cell fluorescence microscopy. A key question raised is whether the factors encoded by hits from the screen act on LDs as ER-LD bridges are formed. This is difficult to analyze in cells at a steady state because many of these factors have multiple localizations in cells (due to other functions independent of LDs) and ER-LD bridge formation appears to occur rapidly, but not synchronously³.

However, we overcame these challenges by developing an assay that allows rapid restoration of a bridge factor in cells depleted of it. For instance, after we deplete cells of

Tango1, we can rescue the GPAT4 targeting defect by adding back Tango1 from a wildtype cell (which expresses endogenous Tango1) by cell-cell fusion (Fig. 5c-f and Supplementary Video 2). In these experiments, ER-to-LD targeting is restored rapidly on select LDs as visualized by GPAT4 targeting. As we now show, this occurs specifically on LDs that are marked by a focus of the fluorescently tagged Sec23, COPII component at ER exit site and another screen hit that we verified. A reticular GPAT4 signal is observed to connect to LDs at these foci, indicating apparent ER-LD connections. Albeit correlative, the temporal sequence of events suggests to us that Sec23 acts at the interface of ER and LDs to establish a bridge that allows for GPAT4 targeting. An additional example is provided in Supplementary Video 3.

A second key question is whether ER-LD bridges are independent of seipin, which many laboratories (including ours) have shown to reside at the neck of forming LDs, maintaining a physical connectivity of ER and LDs. Our prior experiments already showed that seipin-marked connections are not permissible for late ER-to-LD protein targeting (these proteins target later in LD formation due to a restriction by seipin, as shown in Fig. 6b). Therefore, the key question was now whether there exist other ER-LD bridges that mediate late ER-to-LD protein targeting besides seipin-marked ER-LD connections. To address this question, we performed fluorescence recovery after photobleaching (FRAP) experiments at intermediate time points of GPAT4 targeting with some of the protein in the ER and some on LDs. In this experiment, we photobleached fluorescent GPAT4 signal on LDs and observed how the fluorescence is recovered in cells that express fluorescently tagged seipin (tagged at its endogenous locus by genome engineering). These new data show that there are additional ER-LD bridges that are independent of seipin that mediate fluorescence recovery of GPAT4 on LDs (Fig. 6a and Supplementary Videos 4&5).

Additional comments

1) A large number of genes were identified as regulators of GPAT4 targeting (1,100 genes). The rationale for selecting the handful of membrane / trafficking related proteins is clear, but it seems like additional discussion of the other 1,000 genes is warranted. Do 1,100 genes really influence GPAT4 trafficking? What portion are false positives? Are genes that influence lipid droplet size and number controlled for?

We thank the reviewer for bringing up this important point. In brief, the knockdown of 910 genes resulted in significantly lower targeting ratios (robust Z-score < -2.5), excluding ribosomal, proteasomal, spliceosomal genes and RNAi against non-protein coding regions. Our image quantitation pipeline cannot reliably calculate LD targeting ratios when LDs are very small, so we excluded genes that lead to absolute LD areas smaller than that of ACSL RNAi (enzyme involved in TG synthesis, knockdown that causes very small LDs), for instance eliminating *mdy* (DGAT1) or the homologue of CHP1 (an

18activator of GPAT4), which resulted in 751 gene hits. We also excluded genes of which knockdown significantly reduced cell survivability to the extreme (cell count robust Z-score < -2.5), which further reduced the hits to 670 genes. Finally, we focused on those hits that were reproducible between two replicate experiments, which resulted in 302 genes selected as hits. Among these there are about 40 hits with currently unknown functions (and a focus of future studies). Gene Ontology analysis showed that genes involved in vesicle-mediated trafficking were the most strongly enriched (Regulation of ER to Golgi vesicle-mediated transport, p-value = 5.71×10^{-5} ; vesicle cargo loading, p-value = 3.13×10^{-2}), which corroborates the protein complex enrichment analysis of the screen hits (Supplementary Fig. 1f). Other significantly enriched biological process terms included “cell cycle, division, morphogenesis” and “cellular protein metabolic process”, processes that may indirectly affect GPAT4 targeting to LDs. Supplementary Table 1 now includes the Gene Ontology analysis results, as well as the list of unclassified/unmapped genes, and the text was edited to reflect these additional analyses. In addition, all screen data are accessible and can be searched in the LD knowledge portal (<https://lipiddroplet.org/>). We clarified our hit selection and expanded our comments on other screen hits in the revised manuscript.

2) *Do the newly identified late class I lipid droplet proteins show similar temporal kinetics of lipid droplet trafficking?*

Yes, the three newly identified and validated late targeting proteins (LPCAT, ACSL5, DHR57B) target late during LD formation, similar to GPAT4 (Supplementary Fig. 8d,e).

3) *Where is GPAT4 localizing when it can't traffic to the LD? GPAT4 appears to be co-localizing at the ER in Figure 1a, but it is unclear in some panels showing microscopy of GPAT4 in the knockdowns. For example, there is barely any GPAT4 signal in Fig 3a. Is it still ER localized or does it fail to insert into the ER?*

We thank the reviewer for the comment and have performed additional experiments with cells expressing fluorescently tagged, functional GPAT4 tagged at its endogenous locus. RNAi-mediated depletion of Sar1, Sec16, Tango1, Trs20, Rab1, Rint1, Syx5, and Bet1 abolished localization of GPAT4 on LDs and instead lead to GPAT4 co-localizing with ER membrane marker Sec61b (shown below) and ER luminal marker KDEL (Supplementary Fig. 3a). Additionally, we observed a reticular GPAT4 pattern (suggesting ER localization) in EGFP-GPAT4 overexpressing cells depleted of Rab1 or Tango1 (used for the FRAP experiment; Supplementary Fig. 4e and Supplementary Video 1). ER localization of endogenous GPAT4 in cells depleted for these factors is further supported by cellular fractionation data (Supplementary Fig. 5a and Fig. 4d).

4) *It is unclear if these are true knockdowns (KDs) since there is no validation by Western or any other method. It would be nice to see KD validation for their top hits.*

We agree with the reviewer and have added results from three lines of investigation to test the efficiency and specificity of RNAi-mediated depletion. Specifically, we added extensive qPCR data after depletion of key hits and non-hits (Sec12, Sar1, Sec16, Sec23, Tango1, GCC185, Golgin245, Rab1, Rab8, Rab18, Trs20, Trs23, Bet5, Trs120, Rint1, Zw10, Syx5, Syx13, Syx18, Membrin, Sec20, Bet1, Use1, Ykt6, and Sec22; Supplementary Fig. 2b). In addition, mass spectrometry analysis shows depletion of the cognate proteins in RNAi experiments (Supplementary Fig. 8c). We also performed more experiments for phenotype rescue after knock-down (Sec12, Sar1, Sec23, Rab1, Rint1, Syx5, and Bet1; Supplementary Fig. 3b) that support the specificity of knockdowns.

Reviewer #2:

Remarks to the Author:

This study has investigated the pathways by which proteins populate lipid droplets (LDs). Earlier work has established that there are multiple routes, including direct cytosol-to-LD (CYTOLD) and ER-to-LD (ERTOLD) pathways. It was also suggested based on kinetic analysis that the ERTOLD substrates arrive at LDs at different times. The current work provides considerable clarity to the pathways of ERTOLD. The authors document nicely that there is an early and a late ERTOLD pathway that differ in their requirements. Using a screening strategy, they go on to find a set of genes required for late ERTOLD, place these into a sensible working framework, and show how this pathway is different from the early ERTOLD pathway. Strikingly, it emerges that the early ERTOLD pathway restricts certain proteins from going to LDs, and these are later targeted via the late pathway. Removing the restriction factor (seipin) now allows the late clients to enter LDs early, and importantly, renders the genes of the late pathway dispensable for LD targeting.

The work is terrific in every way: clear and convincing data, well-written, biologically important, and sensibly interpreted and discussed. It opens up numerous mechanistic questions about the nature of ER exit sites and how they might link up to LDs, and what each of the late ERTOLD pathway factors contribute to the process. I strongly support its publication and have only a few minor suggestions for improvement. Note that any experimental suggestions in my comments below should be considered optional, and I ask only out of curiosity.

We thank the reviewer for her/his insightful review and strong support for publication. We agree that this work will open many new avenues, and we look forward to more discoveries. Below we provide response to the points raised by the reviewer and outline the new data we added.

Minor comments:

1) Perhaps the most intriguing question that might be worth a couple of lines in the Discussion is why the cell would want to temporally segregate LD clients into early and late. In other words, is there a logic to why early clients would be needed first (or conversely, why late clients should be restricted until later)? Alternatively, is there some shared feature of late clients that make them unsuitable for the early pathway? I feel readers of this study might want to know what the authors are thinking.

These are great points, and we added a consideration of function of early and late targeting pathways to the discussion. For instance, we imagine that early ER-to-LD

targeting proteins may help efficient LD budding from the ER (e.g., via hairpin features that affect surface tension or curvature), and uncontrolled protein targeting during LD formation may adversely alter surface properties, such as surface tension, and disrupt normal LD formation process, as is observed in seipin-deficient cells. On the other hand, once LDs are formed, the late targeting pathway may provide enzymes for LD expansion or remodeling, including remodeling of proteins that target the LD surface. We identified several proteins from the cytosol that target to LDs late after formation (e.g., MLX, spartin, CCT1) perhaps due to such remodeling. We expanded the Discussion to mention possible functions of early vs. late protein targeting pathways from the ER to LDs.

2) It seems to me that forming a bridge between the ERES and LD for the late pathway might depend on an LD protein that is already there. An attractive idea is if an early ERTOLD client actually participates in the late ERTOLD pathway. Is there any evidence for this, perhaps in earlier observations? For example, does eliminating any early clients impact late ERTOLD clients?

This is a very cool idea, and we thank the reviewer for suggesting this interesting possibility. We re-analyzed our data to address this point. In our screen, depletion of 21 total confirmed or presumed early ER-to-LD targeting proteins did not impair GPAT4 targeting to LDs (see also LD knowledge portal: <https://lipiddroplet.org/>). A notable exception for this is AGPAT3. Further studies will be required to determine if this is an effect on LD targeting per se, or a secondary consequence of the LD phenotype associated with AGPAT3 depletion (very small LDs).

3) It is stated that endogenous GPAT4 is decreased in the LD fraction and increased in the microsomal fraction “for each of these factor knockdowns” (referring to Trs20, Rab1, Rint1, Syx5 or Bet1). The decrease in LD is clear. The increase in the membrane fraction in Fig. S3a is less clear and is statistically significant for only Rab1 and Syx5. Perhaps re-phrase or clarify.

We thank the reviewer and have edited the text to clarify this point.

4) In some of the knockdowns that impact LD localisation, the non-localised protein shows some bright puncta (e.g., Rint1 in Fig. 3D). Perhaps a comment about this is worthwhile?

We agree with the reviewer and, in response, performed additional localization studies. To better understand the localization of Rint1 puncta in relation to ER and LDs, we analyzed its localization in comparison to Rab1 signal on LDs and ER marker KDEL. By performing 3D reconstruction of z-stack confocal images, we showed that Rint1 occupies the space between LD surface (marked by Rab1, see point #5 below) and the ER (marked by KDEL) (Fig. 3g,h and Supplementary Fig. 6b,c). This is in agreement with the prior finding that Rint1 functions as a tether between ER and LD membranes⁴.

We also showed that Sec16 overexpression increases association of Tango1 and Sec23 with LDs and quantify the images to better illustrate this point (Fig. 4f-h). This suggests that Sec16 may act upstream of Tango1 and Sec23 and recruit them to LD surface.

5) It is a little hard to tell in Fig. 3F whether the overexpressed Rab1 is really on the LDs or just occupying the space between them. If available, an image with a different overexpressed Rab protein might be helpful for comparison, and would also help convince the reader about the specificity. This is not essential as there is separate evidence for Rab1 being involved; so an optional suggestion for the authors to consider.

We thank the reviewer for this comment. To test if Rab1 is on LDs, we compared its localization with the location of overexpressed Rab18, a protein on LD surfaces⁵. The overexpressed, fluorescently tagged Rab1 and Rab18 showed very robust co-localization around LDs (Fig. 3f, Pearson's correlation coefficient = 0.85; see also the cytofluorograms in Supplementary Fig. 6d). Their respective overlap with the surrounding ER signal was significant but much less robust (R for Rab1 & KDEL = 0.58; R for Rab18 & KDEL = 0.52), indicating that Rab1 and Rab18 co-localize outside the ER, likely on LDs.

Reviewer #3:

Remarks to the Author:

The manuscript of Song et al addresses the targeting pathways of membrane proteins from the ER to lipid droplets (LDs). LDs are formed and mature from the ER membrane and require several metabolic and structural proteins for their function as dynamic fatty acid and lipid depots. How these proteins traffic to and from the LD surface remains incompletely understood. The authors performed a genome-wide screen in fly cells expressing a reporter ER membrane protein that is known to accumulate on LDs with slow kinetics. The screen identified an enrichment of membrane trafficking components. In follow-up analyses, the authors validated the requirement of these components such as Rab1, Snares, a TRAPP subunit and ER exit site machinery and further showed that removal of seipin rescues these targeting defects, consistent with a role of seipin in controlling traffic at ER-LD contact sites. Finally, membrane proteins that use this ER to LD targeting were identified.

Overall, the screen and follow-up experiments are well executed (but see below for some limitations on imaging/validation) and I find the topic very interesting. The paper is well written. The authors demonstrate a clear requirement for certain trafficking factors in the protein targeting from the ER to LDs. However, my view is that the study does not significantly extend beyond reporting and characterizing the screen hits. Previous studies, including from the authors, have established a link between trafficking factors and LDs but the underlying basis remains unclear. It would be important for the authors to at least test their main hypothesis of a membrane fusion reaction at the LD site required for ER to LD traffic.

We thank the reviewer for this comment and agree. We have now added substantial new mechanistic data that strengthen our conclusions.

Our laboratory previously reported ER-LD bridges by electron tomography. It therefore is logical to ask whether these bridges depend on the factors we identify. Although we performed extensive EM experiments to address this, we encountered three confounding factors that make this experiment next to impossible. i) Although abundant on a subset of LDs³, bridges are still rare in electron micrographs of EM sections (because they only sample a very small section of a small percentage of the LDs), thereby complicating the quantitation of their absence. ii) Even in the absence of bridge factors, seipin-mediated ER-LD connections persist (Supplementary Fig. 8b**), thus complicating analysis by EM. iii) Electron tomography, which is better suited to visualize bridges, does not allow the collection of enough data to make a statistically sound argument. We edited the text to better explain these issues.**

An alternative approach is to visualize the ER-LD bridges with live-cell fluorescence microscopy. A key question raised is whether the factors encoded by hits from the screen act on LDs as ER-LD bridges are formed. This is difficult to analyze in cells at a steady state because many of these factors have multiple localizations in cells (due to other functions independent of LDs) and ER-LD bridge formation appears to occur rapidly, but not synchronously³.

However, we overcame these challenges by developing an assay that allows rapid restoration of a bridge factor in cells depleted of it. For instance, after we deplete cells of Tango1, we can rescue the GPAT4 targeting defect by adding back Tango1 from a wildtype cell (which expresses endogenous Tango1) by cell-cell fusion (Fig. 5c-f and Supplementary Video 2). In these experiments, ER-to-LD targeting is restored rapidly on select LDs as visualized by GPAT4 targeting. As we now show, this occurs specifically on LDs that are marked by a focus of the fluorescently tagged Sec23, COPII component at ER exit site and another screen hit that we verified. A reticular GPAT4 signal is observed to connect to LDs at these foci, indicating apparent ER-LD connections. Albeit correlative, the temporal sequence of events suggests to us that Sec23 acts at the interface of ER and LDs to establish a bridge that allows for GPAT4 targeting. An additional example is provided in Supplementary Video 3.

A second key question is whether ER-LD bridges are independent of seipin, which many laboratories (including ours) have shown to reside at the neck of forming LDs, maintaining a physical connectivity of ER and LDs. Our prior experiments already showed that seipin-marked connections are not permissible for late ER-to-LD protein targeting (these proteins target later in LD formation due to a restriction by seipin, as shown in Fig. 6b). Therefore, the key question was now whether there exist other ER-LD bridges that mediate late ER-to-LD protein targeting besides seipin-marked ER-LD connections. To address this question, we performed fluorescence recovery after photobleaching (FRAP) experiments at intermediate time points of GPAT4 targeting with some of the protein in the ER and some on LDs. In this experiment, we photobleached fluorescent GPAT4 signal on LDs and observed how the fluorescence is recovered in cells that express fluorescently tagged seipin (tagged at its endogenous locus by genome engineering). These new data show that there are additional ER-LD bridges that are independent of seipin that mediate fluorescence recovery of GPAT4 on LDs (Fig. 6a and Supplementary Videos 4&5).

It would be also important to confirm that inhibition of ER to Golgi traffic, which can affect ER membrane, does not have an indirect impact on GPAT4 movement, especially under conditions of fatty acid overload.

Thank you for this suggestion. We agree with the referee and addressed this point experimentally. Specifically, we used FRAP to test the mobility of GPAT4 in the ER, in

26cells depleted of late ER-to-LD protein targeting factors with or without excess fatty acid treatment. In these experiments (Supplementary Fig. 4c-e and Supplementary Video 1), the tau values among various knockdown conditions (RNAi against Sar1, Tango1, Rab1, Rint1, or Syx5) remained comparable to the control (RNAi against LacZ), indicating that there is no difference in GPAT4 mobility. Additionally, absence of seipin allows targeting of late cargoes even in the absence of these factors (Fig. 6c,d). This indicates that depletion of late ER-to-LD protein targeting machinery does not impair the ability of the cargoes to move to LDs but instead abolishes their path to LDs, and that the absence of seipin provides an alternative route for late ER-to-LD targeting.

We also compared our screen results with secretory pathway screen also performed in *Drosophila* cells⁶. In the secretory pathway screen, *Drosophila* S2 cells engineered to secrete HRP were subjected to a genome-scale library of dsRNA, and the effect of gene knockdowns on secretion was measured by analyzing for the peroxidase activity in the medium using chemiluminescence. As expected, RNAi of many of our hits (Tango1, Sec23, Rab1, Rint1, Syx5, Bet1) significantly reduced HRP secretion (Supplementary Fig. 1g). Nevertheless, hits of the two screens were poorly correlated ($R = 0.3785$), such that RNAi of many of the secretion screen hits (Rab11, Tango10, Tango5, C11orf54, CG4835) did not reduce GPAT4 targeting to LDs. This likely suggests that GPAT4 targeting phenotype found in our screen cannot solely be explained by the defect in the secretion pathway.

Finally, it is not clear to me why membrane bridges connecting the GPAT4-containing LDs and the ER would necessarily need a membrane fusion event as opposed to structural rearrangement and/or tubulation of the ER membrane.

This is an interesting point. Our model includes that the seipin ER-LD bridge restricts specific ER proteins from accessing LDs. Therefore, some type of additional fusion between the organelles would be required to establish a physical connectivity for membrane-embedded proteins to migrate between the organelles. An interesting alternative would be that the ER-LD connectivity at seipin foci is re-modelled. Although we cannot exclude this model completely, we provide additional data that ER-LD bridges are formed independently of seipin foci, making this possibility less likely. Additionally, data from our lab and others suggest that there is generally one seipin focus per LD^{7,8}, whereas we find large LDs in S2 cells have many ER-LD connections³. We edited the manuscript Discussion to clarify this point.

Other comments:

An argument for the specificity of the pathway is the localization of membrane fusion factors and ER exit site machinery to LDs. I think these results could be strengthened. For example for the localization of Rab1 and Rint1 to LDs (Fig3f), one wonders whether the distribution of Rint1

corresponds to an ER subdomain in the vicinity of LDs rather than LDs. This also applies to the Tango/Sec16 localization (Fig. 4e, f,g). Higher resolution methods would be beneficial here. Alternatively, how would these localizations compare to other ER resident and trafficking components following the addition of oleate?

We agree with the referee and performed additional experiments to strengthen this line of investigation.

Since Rab18 (ER-Golgi trafficking factor that was not required for late ER to LD protein targeting) localizes to LD surfaces by immunogold labeling in electron microscopy⁵, we performed co-localization analysis of overexpressed, tagged Rab1 with Rab18. Consistent with prior biochemical data supporting Rab1 association with LDs, Rab1 co-localized strongly with Rab18 on LDs (Fig. 3f, Pearson's correlation coefficient = 0.85; see also the cytofluorograms in Supplementary Fig. 6d). Their respective co-localization with the ER marker Halo-KDEL was significantly less (R for Rab1 & KDEL = 0.58; R for Rab18 & KDEL = 0.52), indicating that Rab1 and Rab18 co-localize outside the ER, likely on LDs.

To better understand the distribution of Rint1 puncta in relation to the ER and LDs, we performed 3D reconstruction of the z-stack images and found that Rint1 occupies the space between the LD surface (marked by Rab1) and the ER (marked by KDEL) (Fig. 3g,h and Supplementary Fig. 6b,c). This is consistent with the prior report that Rint1 functions as a tether between ER and LD membranes⁴. Unlike Rint1, which strongly enriched around LDs, Zw10 (a membrane trafficking factor known to form a tethering complex with Rint1) remained cytosolic (Supplementary Fig. 6f), further supporting the specific requirement of Rint1 in ER-to-LD protein targeting.

These results have been added to the revised manuscript.

Most of the membrane trafficking hits from the screen that have been followed up have not been validated by confirming their protein levels - could be this a reason as to why components of the same trafficking complexes show different phenotypes? Also, for positive hits with the exception of Rab1, there are no rescue experiments.

Again, we agree with the referee and, in response, added new experimental data. We tested the effect of RNAi-mediated expression changes, including for trafficking factors that are and are not required for late ER-to-LD protein targeting (Supplementary Fig. 2b; required for targeting: Sec12, Sar1, Sec16, Sec23, Tango1, Rab1, Trs20, Rint1, Syx5, Membrin, Bet1, Ykt6; not required for targeting: GCC185, Golgin245, Rab8, Rab18, Trs23, Bet5, Trs120, Zw10, Syx13, Syx18, Sec20, Use1, Sec22). In addition, proteomic data confirms that target proteins are specifically depleted from cells upon RNAi (Supplementary Fig. 8c). Moreover, we added new rescue experiments that test and

28confirm the specificity of the phenotypes to the depletion of membrane bridge factors (Supplementary Fig. 3b; Sec12, Sar1, Sec23, Rab1, Rint1, Syx5, and Bet1).

*I am not sure I understand the statement of a machinery that "...supports *heterotypic* organelle fusion of the ER and LDs" in the introduction and discussion. LDs are derived from the ER and essentially constitute a specialized ER domain rather than an organelle with a separate identity.*

We thank the reviewer for this comment. In this context, "heterotypic" refers to new additional connections between a bilayer membrane and the LD monolayer membrane. We removed the term and clarified this point in the revised manuscript.

REFERENCES

1. Kory, N., Farese, R. V. & Walther, T. C. Targeting Fat: Mechanisms of Protein Localization to Lipid Droplets. *Trends in Cell Biology* **26**, 535–546 (2016).
2. Olarte, M. J., Swanson, J. M. J., Walther, T. C. & Farese, R. V. The CYTOLD and ERTOLD pathways for lipid droplet–protein targeting. *Trends in Biochemical Sciences* vol. 47 39–51 (2022).
3. Wilfling, F. *et al.* Triacylglycerol synthesis enzymes mediate lipid droplet growth by relocating from the ER to lipid droplets. *Developmental Cell* **24**, 384–399 (2013).
4. Xu, D. *et al.* Rab18 promotes lipid droplet (LD) growth by tethering the ER to LDs through SNARE and NRZ interactions. *Journal of Cell Biology* **217**, 975–995 (2018).
5. Liu, P. *et al.* Rab-regulated interaction of early endosomes with lipid droplets. *Biochimica et Biophysica Acta* **1773**, 784–793 (2007).
6. Bard, F. *et al.* Functional genomics reveals genes involved in protein secretion and Golgi organization. *Nature* **439**, 604–607 (2006).
7. Wang, H. *et al.* Seipin is required for converting nascent to mature lipid droplets. *Elife* **5**, (2016).
8. Salo, V. T. *et al.* Seipin regulates ER – lipid droplet contacts and cargo delivery. 1–18 (2016).

Decision Letter, first revision:

Subject: Your manuscript, NCB-S46664A
Message: Our ref: NCB-S46664A

19th May 2022

Dear Tobi,

Thank you for submitting your revised manuscript "Identification of two pathways mediating protein targeting from ER to lipid droplets" (NCB-S46664A) and thank you for your thorough efforts to strengthen the evidence supporting the two pathways and model. The revision has now been seen by two of the original referees and their comments are below. Reviewer #1 agreed to assess your responses to Rev#2. The reviewers both found that the paper has improved in revision, and Rev#1 indicated in comments to us that they felt your responses to Rev#2's points were also compelling. Therefore, we'll be happy in principle to publish the study in Nature Cell Biology, pending minor revisions to satisfy the referees' final requests and to comply with our editorial and formatting guidelines.

****The current version of your manuscript is in a PDF format; could you please email us a copy of the file in an editable format (Microsoft Word or LaTeX)? With apologies, we can not proceed with PDFs at this stage.****

Once we have the editable Word file for the manuscript, we will start performing detailed checks on your paper and will send you a checklist detailing our editorial and formatting requirements in about ~a week/10 days. Please do not upload the final materials and make any revisions until you receive this additional information from us.

Thank you again for your interest in Nature Cell Biology. Please do not hesitate to contact me if you have any questions.

Sincerely,

Melina

Melina Casadio, PhD

30Senior Editor, Nature Cell Biology
ORCID ID: <https://orcid.org/0000-0003-2389-2243>

Reviewer #1 (Remarks to the Author):

The authors have thoughtfully and thoroughly responded to my previous comments. I recommend publication and commend the authors on a beautiful and exciting manuscript!

Reviewer #3 (Remarks to the Author):

The authors have addressed my major concerns. I find that the manuscript is significantly improved. I only have a minor query: in supplementary Fig. 6b, please indicate where do the inlays at the bottom come from.

Decision Letter, final requests:

Subject: NCB: Your manuscript, NCB-S46664A
Message: Our ref: NCB-S46664A

2nd June 2022

Dear Dr. Walther,

Thank you for your patience as we've prepared the guidelines for final submission of your Nature Cell Biology manuscript, "Identification of two pathways mediating protein targeting from ER to lipid droplets" (NCB-S46664A). Please carefully follow the step-by-step instructions provided in the attached file, and add a response in each row of the table to indicate the changes that you have made. Please also check and comment on any additional marked-up edits we have proposed within the text. Ensuring that each point is addressed will help to ensure that your revised manuscript can be swiftly handed over to our production team.

We would like to start working on your revised paper, with all of the requested files and forms, as soon as possible (preferably within one week). Please get in contact with us if you anticipate delays.

31When you upload your final materials, please include a point-by-point response to any remaining reviewer comments.

In recognition of the time and expertise our reviewers provide to Nature Cell Biology's editorial process, we would like to formally acknowledge their contribution to the external peer review of your manuscript entitled "Identification of two pathways mediating protein targeting from ER to lipid droplets". For those reviewers who give their assent, we will be publishing their names alongside the published article.

Nature Cell Biology offers a Transparent Peer Review option for new original research manuscripts submitted after December 1st, 2019. As part of this initiative, we encourage our authors to support increased transparency into the peer review process by agreeing to have the reviewer comments, author rebuttal letters, and editorial decision letters published as a Supplementary item. When you submit your final files please clearly state in your cover letter whether or not you would like to participate in this initiative. Please note that failure to state your preference will result in delays in accepting your manuscript for publication.

Cover suggestions

As you prepare your final files we encourage you to consider whether you have any images or illustrations that may be appropriate for use on the cover of Nature Cell Biology.

Nature Cell Biology has now transitioned to a unified Rights Collection system which will allow our Author Services team to quickly and easily collect the rights and permissions required to publish your work. Approximately 10 days after your paper is formally accepted, you will receive an email in providing you with a link to complete the grant of rights. If your paper is eligible for Open Access, our Author Services team will also be in touch regarding any additional information that may be required to arrange payment for your article.

Please note that Nature Cell Biology is a Transformative Journal (TJ). Authors may publish their research with us through the traditional subscription access route or make their paper immediately open access through payment of an article-processing charge (APC). Authors will not be required to make a final decision about access to their article until it has been accepted. Find out more about Transformative Journals

Authors may need to take specific actions to achieve compliance with funder and institutional open access mandates. If your research is supported by a funder that requires immediate open access (e.g. according to Plan S principles) then you should select the gold OA route, and we will direct you to the compliant route where possible. For authors selecting the subscription publication route, the journal's standard licensing terms will need to be accepted, including self-archiving policies. Those licensing terms will supersede any other terms that the author or any third party may assert apply to any version of the manuscript.

For information regarding our different publishing models please see our Transformative Journals page. If you have any questions about costs, Open Access requirements, or our legal forms, please contact ASJournals@springernature.com.

[REDACTED]

Best regards,

Nyx Hills
Staff
Nature Cell Biology

On behalf of

Melina Casadio, PhD
Senior Editor, Nature Cell Biology
ORCID ID: <https://orcid.org/0000-0003-2389-2243>

Reviewer #1:

Remarks to the Author:

The authors have thoughtfully and thoroughly responded to my previous comments. I recommend publication and commend the authors on a beautiful and exciting manuscript!

Reviewer #3:

Remarks to the Author:

The authors have addressed my major concerns. I find that the manuscript is significantly improved. I only have a minor query: in supplementary Fig. 6b, please indicate where do the inlays at the bottom come from.

Final Decision Letter:

Dear Dr Walther,

I am pleased to inform you that your manuscript, "Identification of two pathways mediating protein

34targeting from ER to lipid droplets", has now been accepted for publication in Nature Cell Biology. Congratulations on this very nice study!

Please note that *Nature Cell Biology* is a Transformative Journal (TJ). Authors may publish their research with us through the traditional subscription access route or make their paper immediately open access through payment of an article-processing charge (APC). Authors will not be required to make a final decision about access to their article until it has been accepted. Find out more about Transformative Journals

Authors may need to take specific actions to achieve compliance with funder and institutional open access mandates. If your research is supported by a funder that requires immediate open access (e.g. according to Plan S principles) then you should select the gold OA route, and we will direct you to the compliant route where possible. For authors selecting the subscription

35publication route, the journal's standard licensing terms will need to be accepted, including self-archiving policies. Those licensing terms will supersede any other terms that the author or any third party may assert apply to any version of the manuscript.

If you have not already done so, we strongly recommend that you upload the step-by-step protocols used in this manuscript to the Protocol Exchange (www.nature.com/protocolexchange), an open online resource established by Nature Protocols that allows researchers to share their detailed experimental know-how. All uploaded protocols are made freely available, assigned DOIs for ease of citation and are fully searchable through nature.com. Protocols and Nature Portfolio journal papers in which they are used can be linked to one another, and this link is clearly and prominently visible in the online versions of both papers. Authors who performed the specific experiments can act as primary authors for the Protocol as they will be best placed to share the methodology details, but the Corresponding Author of the present research paper should be included as one of the authors. By uploading your Protocols to Protocol Exchange, you are enabling researchers to more readily reproduce or adapt the methodology you use, as well as increasing the visibility of your protocols and papers. You can also establish a dedicated page to collect your lab Protocols. Further information can be found at www.nature.com/protocolexchange/about

With kind regards,

Melina

Melina Casadio, PhD
Senior Editor, Nature Cell Biology
ORCID ID: <https://orcid.org/0000-0003-2389-2243>

** Visit the Springer Nature Editorial and Publishing website at www.springernature.com/editorial-and-publishing-jobs for more information about our career opportunities. If you have any questions please click here.**